# Early myelination involves the dynamic and repetitive ensheathment of axons which resolves through a low and consistent stabilization rate

Adam R Almeida, Wendy B Macklin*

Department of Cell and Developmental Biology, University of Colorado School of Medicine, Aurora, United States

*For correspondence:
wendy.macklin@cuanschutz.edu

Competing interest: The authors declare that no competing interests exist.

**Abstract** Oligodendrocytes in the central nervous system exhibit significant variability in the number of myelin sheaths that are supported by each cell, ranging from 1 to 50 (1-8). Myelin production during development is dynamic and involves both sheath formation and loss (3, 9-13). However, how these parameters are balanced to generate this heterogeneity in sheath number has not been thoroughly investigated. To explore this question, we combined extensive time-lapse and longitudinal imaging of oligodendrocytes in the developing zebrafish spinal cord to quantify sheath initiation and loss. Surprisingly, we found that oligodendrocytes repetitively ensheathed the same axons multiple times before any stable sheaths were formed. Importantly, this repetitive ensheathment was independent of neuronal activity. At the level of individual oligodendrocytes, each cell initiated a highly variable number of total ensheathments. However, ~80–90% of these ensheathments always disappeared, an unexpectedly high, but consistent rate of loss. The dynamics of this process indicated rapid membrane turnover as ensheathments were formed and lost repetitively on each axon. To better understand how these sheath initiation dynamics contribute to sheath accumulation and stabilization, we disrupted membrane recycling by expressing a dominant-negative mutant form of Rab5. Oligodendrocytes over-expressing this mutant did not show a change in early sheath initiation dynamics but did lose a higher percentage of ensheathments in the later stabilization phase. Overall, oligodendrocyte sheath number is heterogeneous because each cell repetitively initiates a variable number of total ensheathments that are resolved through a consistent stabilization rate.

## Editor's evaluation

The study by Almeida and Macklin takes advantage of the power of zebrafish as a model system to study the dynamic behavior of myelinating oligodendrocytes in a living system. Using a combination of time-lapse imaging and transgenic constructs in larval zebrafish, the authors compellingly demonstrate that oligodendrocytes balance sampling of axons with stabilization of myelin sheaths at a stereotypical rate during development. The study provides valuable new insights into how oligodendrocytes undertake myelination and will be useful to the field.

## Introduction

Myelin sheaths are specialized cellular structures produced by oligodendrocytes in the central nervous system (CNS) that wrap around axons to accelerate the velocity of action potential conduction and to provide trophic support to neurons (reviewed in *Almeida and Lyons, 2017*; *Stadelmann et al., 2019*).

The number of myelin sheaths that are supported by each oligodendrocyte is remarkably variable (reviewed in *Simons and Nave, 2015*; *Williamson and Lyons, 2018*). Experiments using genetically encoded fluorescent reporters and immunostaining in both murine and zebrafish models establish that oligodendrocytes have anywhere from one sheath per cell up to about 50 (*Almeida et al., 2011*; *Chong et al., 2012*; *Czopka et al., 2013*; *Bechler et al., 2015*; *Osanai et al., 2017*; *Hughes et al., 2018*; *Bacmeister et al., 2020*; *Swire et al., 2019*). However, the importance of this heterogeneity in sheath number and how it is created is still unclear.

Time-lapse imaging in zebrafish has shown that the dynamics of developmental myelination include both sheath production and sheath loss (*Czopka et al., 2013*; *Liu et al., 2013*; *Hines et al., 2015*; *Mensch et al., 2015*; *Hughes and Appel, 2020*; *Djannatian et al., 2023*). As past studies have not thoroughly compared sheath formation and loss across the same individual cells, the question remains as to the relative contribution of these two mechanisms to establishing oligodendrocyte sheath number. Understanding this balance is important because there are dramatically different cellular and molecular mechanisms that regulate sheath formation and loss. Thus, both processes could potentially be manipulated to improve myelin regeneration.

To address this gap in our knowledge we combined extensive time-lapse and longitudinal imaging in the developing zebrafish spinal cord to quantify both sheath initiation and loss from the same cells. Unexpectedly, oligodendrocytes repetitively formed and retracted ensheathments from the same axon multiple times before any stable sheaths were formed, a phenomenon that was independent of neuronal activity. These dynamics resulted in all oligodendrocytes consistently stabilizing only ~10–20% of the total number of immature ensheathments formed by each cell. Altering the endocytic recycling pathway in oligodendrocytes did not disrupt early sheath initiation dynamics but resulted in fewer ensheathments being maintained during the later stabilization phase. These studies suggest that oligodendrocyte sheath number is determined by the total number of ensheathments that are formed by each cell. Sheath stabilization is an important part of the process since it occurs at such a low rate but is proportionally regulated across oligodendrocytes.

## Results
### Regional differences in oligodendrocyte sheath number during developmental myelination

The number of sheaths supported by a single oligodendrocyte in the zebrafish spinal cord at 4 days post fertilization (dpf) is widely variable (*Almeida et al., 2011*). However, it is unclear how this variability arises during development. Oligodendrocytes in the spinal cord myelinate either dorsal or ventral axon tracts and occupy these regions with differing cell densities. There are ~2-fold more myelinating oligodendrocytes in the ventral tracts compared to the dorsal tracts (*Figure 1A and B*). These domains have different functions, with ascending dorsal axons primarily transmitting sensory

> **Box 1. Definitions.**
>
> **Sheath initiation:** The beginning steps of wrapping oligodendrocyte membrane around an axon.
> **Ensheathment:** Oligodendrocyte membrane wrapped around an axon. This does not imply knowledge of the composition of the oligodendrocyte membrane.
> **Stabilized ensheathment:** An ensheathment that persists.
> **Destabilized ensheathment:** An ensheathment that disappears and is lost.
> **Repetitive ensheathment:** The iterative wrapping and unwrapping of oligodendrocyte membrane around an axonal domain by the same oligodendrocyte.
> **Axonal domain:** A region of an axon that is ensheathed by an oligodendrocyte.
> **Stabilized axonal domain:** An axonal domain where an ensheathment forms and persists on the axon.
> **Destabilized axonal domain:** An axonal domain where an ensheathment forms but is then lost and does not reform.

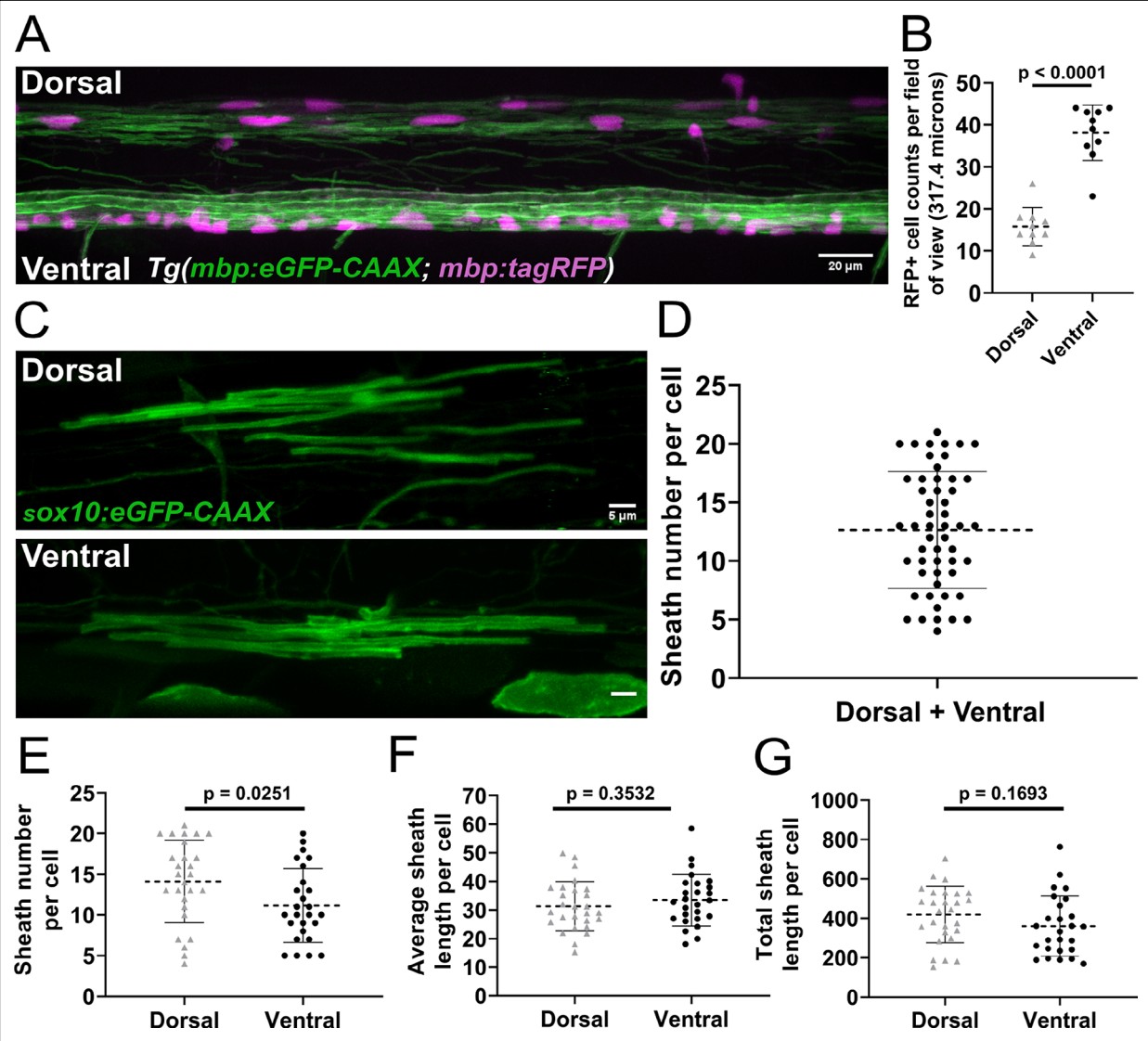

**Figure 1.** Sheath number was highly variable, but oligodendrocytes in the dorsal region of the spinal cord had more sheaths per cell than oligodendrocytes in the ventral region. (**A**) Lateral image of the spinal cord of a living *Tg(mbp:eGFP-CAAX; mbp:TagRFP)* larva (scale bar = 20 µm). (**B**) RFP+ oligodendrocyte cell counts in the dorsal and ventral regions of the spinal cord per field of view (317.4 µm long). (n=10 larvae). (**C**) Representative lateral images of dorsal and ventral oligodendrocytes in the spinal cord of living larvae at 4 days post fertilization (dpf) labeled by *sox10:eGFP-CAAX* (scale bar = 5 µm). (**D**) Sheath number per cell, all data combined (n=53 cells/53 larvae). (**E–G**) Sheath number per cell (**E**), average sheath length per cell (**F**), total sheath length per cell (**G**), comparing the dorsal and ventral regions (dorsal n=27 cells/27 larvae, ventral n=26 cells/26 larvae). The dashed lines in each plot represent average values with all data points shown. The error bars represent standard deviation. Significance determined by Mann-Whitney tests (see associated source data).

The online version of this article includes the following source data for figure 1:

**Source data 1.** Excel spreadsheet with the oligodendrocyte cell counts for *Figure 1B*.

**Source data 2.** Excel spreadsheet with the oligodendrocyte sheath and length numbers for *Figure 1D–G*.

information and descending ventral axons transmitting motor information (*Thau and Reddy, 2022*). We investigated whether some of the variability in oligodendrocyte sheath number comes from quantifying cells from both regions together. Zebrafish embryos were injected at the single-cell stage with a *sox10:eGFP-CAAX* plasmid to sparsely label the membrane of individual oligodendrocytes. At 4 dpf, sheath number ranged from 4 to 21 per cell (*Figure 1C and D*), which is in line with previous work (*Almeida et al., 2011*). Importantly, oligodendrocytes in the dorsal region of the spinal cord had more sheaths on average than oligodendrocytes in the ventral region (*Figure 1E*). No statistically significant

differences in average sheath length or the total sheath length were found when comparing these groups (*Figure 1F and G*). Oligodendrocytes in the ventral spinal cord that myelinated Mauthner axons were excluded from all analyses in this manuscript. To investigate how the regional difference in sheath number is generated, we compared sheath initiation and loss by oligodendrocytes in the dorsal and ventral regions of the spinal cord.

## Oligodendrocytes repetitively ensheathed axons in an activity-independent manner

To establish parameters for quantifying the dynamics of sheath initiation and loss, we sparsely labeled neurons by injecting the pan-neuronal *neuroD:tagRFP-CAAX* axon reporter into *Tg(nkx2.2:eGFP-CAAX)*, a transgenic myelin reporter line. The earliest stages of eGFP-CAAX-labeled sheath formation on tagRFP-CAAX-labeled axons were captured with extensive time-lapse imaging from 2.5 to 3 dpf (*Figure 2A*, see Materials and methods for more details). We could consistently identify immature ensheathments that were ~2 µm or longer, surrounding axons with a cylindrical shape (*Figure 2B and C*). Unexpectedly, each oligodendrocyte repetitively ensheathed the same axonal domain an average of 2–3 times before any stable sheaths were formed (*Figure 2D, E* [T0', T15', T45', T60']). We define repetitive ensheathment as multiple rounds of sheath initiation/loss on the same domain of an axon. Each axon domain was defined as the region between the two most lateral ensheathment attempts made by the same oligodendrocyte during a series of repetitive ensheathments. In cases with only a single ensheathment attempt, the axon domain was considered the region directly underneath the sheath. Of the 25 axonal domains analyzed, 14 of them ended up with a stabilized sheath (*Figure 2F*) and 11 were lost (*Figure 2G*). The total number of ensheathment attempts on each axonal domain ranged from 1 to 6, and 72% of these domains were ensheathed more than once. This repetitive ensheathment phenomenon was therefore common in our experiment and possibly fundamental to the process of building myelin sheaths.

It is important to note that we performed this time-lapse imaging paradigm using tricaine as anesthesia. This drug is a sodium channel blocker that silences action potentials throughout the CNS. Neuronal activity is important for myelination, and it is possible that the repetitive axonal ensheathment that we observed could result from silencing neuronal activity. We tested this using the same axonal ensheathment imaging paradigm (*Figure 3A*). We anesthetized each larva in either tricaine or pancuronium bromide, which is a neuromuscular blocking drug that does not dampen neuronal activity in the CNS (https://www.ncbi.nlm.nih.gov/books/NBK538346/). It has also been used previously in zebrafish activity-dependent myelination studies (*Hughes and Appel, 2020*; *Koudelka et al., 2016*; *Hughes and Appel, 2019*). Importantly, there was essentially no difference in the average number of times an oligodendrocyte ensheathed the same axonal domain when comparing these forms of anesthesia (*Figure 3B–D*). Since we sometimes observed more than one axonal domain on the same axon, we also normalized the data by averaging the total number of ensheathment attempts across all domains of each axon (*Figure 3E*). However, there was still no difference between the two forms of anesthesia. We also found that 43% of the domains were stabilized (16 of 37) at the end of the imaging paradigm for the tricaine group and 55% were stabilized (16 of 29) for the pancuronium bromide group (*Figure 3F*). A Fisher's exact test found no significant difference between these values.

As a follow-up to this work, we performed a quality control experiment to demonstrate that there was a measurable difference in the CNS activity of larvae anesthetized in tricaine vs pancuronium bromide in our experiment. To do this we measured calcium transients in the zebrafish spinal cord using the *Tg(elav3:H2B-GCaMP6f)* pan-neuronal transgenic line (*Dunn et al., 2016*). We anesthetized larvae at 2.5 dpf in pancuronium bromide and imaged the spinal cord above the yolk sac extension at 4 Hz (*Figure 3—figure supplement 1A*). Each larva was then spiked with embryo media containing either tricaine or pancuronium bromide (control) for 5 min. After this, another video was collected on the same cells. There was no significant difference in the number of calcium transients in the pre- and post-videos for the controls, but there was a clear reduction in transients after the tricaine spike (*Figure 3—figure supplement 1B-D*). Due to the variability in the total number of transients that we observed across all the larvae, we also normalized the data by dividing the number of transients in the post-video by the number of transients in the pre-video to generate a percentage. There was still a clear difference in the percentage of calcium transients in the tricaine spiked group relative to the controls (*Figure 3—figure supplement 1E*). Collectively, this experiment allows us to more strongly

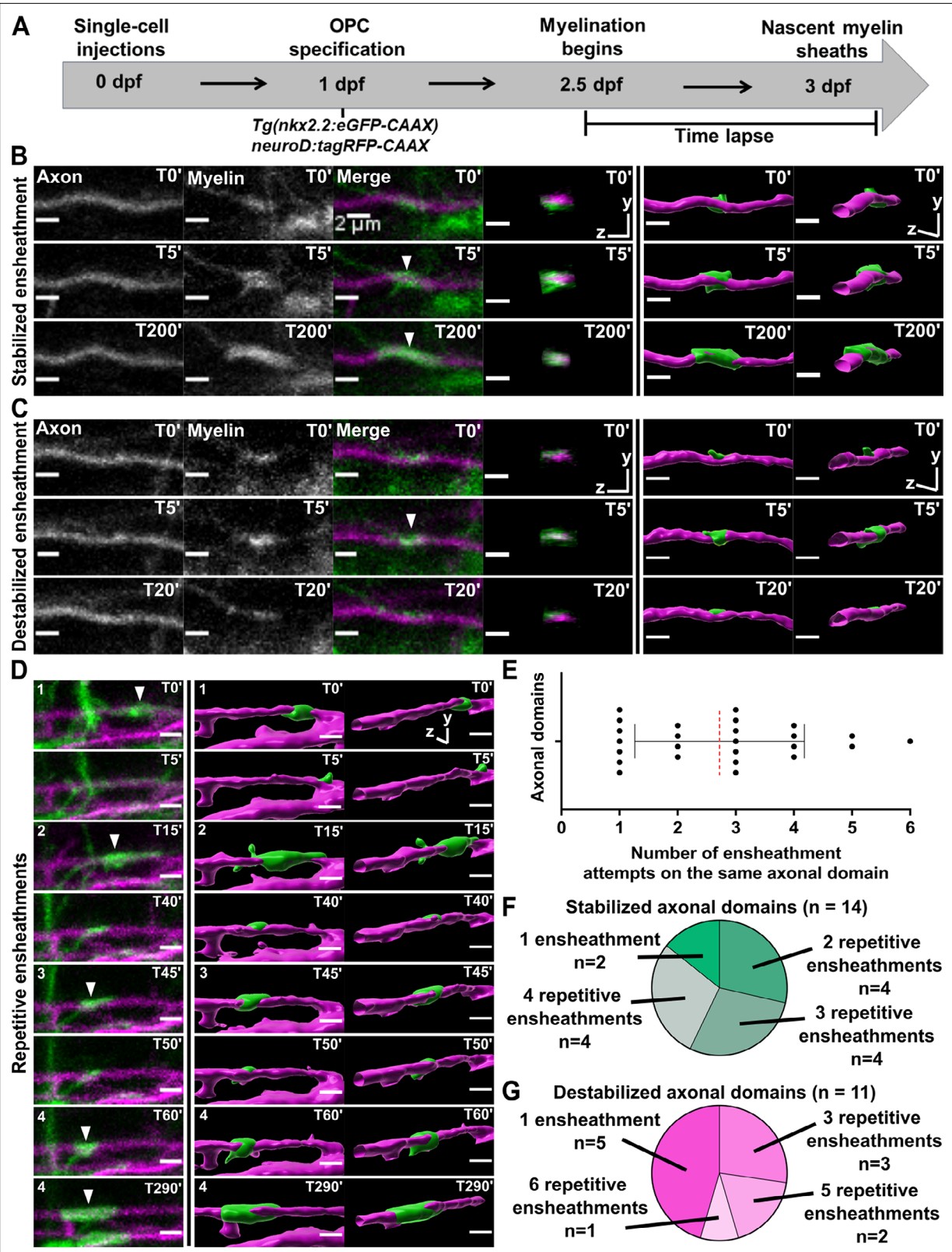

**Figure 2.** Oligodendrocytes repetitively ensheathed the same axonal domains before stable sheaths were formed. (**A**) Axonal ensheathment dynamics experimental paradigm. (**B–C**) Representative lateral images of immature ensheathments forming in the spinal cord of living *Tg(nkx2.2:eGFP-CAAX)* (myelin in green) larvae time-lapsed for 18 hr with a 5 min imaging interval starting at 2.5 days post fertilization (dpf). Axons were labeled with *neuroD:tagRFP-CAAX* (magenta). YZ images are cross sections of the axon (volume projections from Imaris). 3D reconstructions were generated

*Figure 2 continued on next page*

*Figure 2 continued*

in Imaris. (**B**) Time-lapse imaging of a stabilized ensheathment (scale bar = 2 µm). (**C**) Time-lapse imaging of a destabilized ensheathment (scale bar = 2 µm). (**D**) Representative lateral images of an axonal domain that was repetitively ensheathed four times before a sheath was stabilized. 3D reconstructions were generated in Imaris (scale bar = 2 µm). (**E**) Plot showing the number of times each axonal domain was ensheathed, for example, seven axonal domains had only one ensheathment attempt. Dotted red line represents the average number of ensheathment attempts. The error bar represents standard deviation (n=25 axonal domains, 18 axons, 12 larvae, 68 ensheathments imaged total). (**F**) Pie chart of the number of axonal domains with a final stable ensheathment and the number of ensheathment attempts that were made on each of those domains (n=14 axonal domains). (**G**) Pie chart of the number of destabilized axonal domains and the number of ensheathment attempts that were made on each of those domains (n=11 axonal domains) (see associated source data and supplementary video files).

The online version of this article includes the following video and source data for figure 2:

**Source data 1.** Excel spreadsheet with the repetitive ensheathment numbers and the stabilization status for each axonal domain.

**Figure 2—video 1.** *Figure 2B* fluorescence time-lapse video for the stabilized ensheathment example.

https://elifesciences.org/articles/82111/figures#fig2video1

**Figure 2—video 2.** *Figure 2B* 3D reconstruction_360° rotation video for the stabilized ensheathment example.

https://elifesciences.org/articles/82111/figures#fig2video2

**Figure 2—video 3.** *Figure 2C* fluorescence time-lapse video for the destabilized ensheathment example.

https://elifesciences.org/articles/82111/figures#fig2video3

**Figure 2—video 4.** *Figure 2C* 3D reconstruction_360° rotation video for the destabilized ensheathment example.

https://elifesciences.org/articles/82111/figures#fig2video4

**Figure 2—video 5.** *Figure 2D* fluorescence time-lapse video for the repetitive ensheathment example.

https://elifesciences.org/articles/82111/figures#fig2video5

**Figure 2—video 6.** *Figure 2D* 3D reconstruction_360° rotation video for the repetitive ensheathment example.

https://elifesciences.org/articles/82111/figures#fig2video6

conclude that the repetitive ensheathment phenomenon that we observed in *Figures 2 and 3* is independent of neuronal activity.

Further assessment of the two axonal ensheathment experiments from *Figures 2 and 3* indicates that for the tricaine condition, 72.6% of all axonal domains were ensheathed more than once (*Figure 3—figure supplement 2A*). Similarly, 65.5% of domains were ensheathed more than once for the pancuronium bromide group. In 100% of these instances, each axonal domain was repetitively ensheathed by the same oligodendrocyte (*Figure 3—figure supplement 2B*). Additionally, the same cell process performs each ensheathment and seems to maintain continuous contact with the axon ~63–68% of the time (*Figure 3—figure supplement 2C and D*, see Materials and methods for more details). This suggests that maintaining contact with the axon could be important for sheath formation.

We next tried to better understand how the stabilization of sheaths might result from this repetitive ensheathment process. Due to the similarity in the outcomes for the tricaine and pancuronium bromide conditions in the previous paragraph, we combined all our data together to look at other metrics. We first looked at whether the average number of repetitive ensheathments for an axonal domain is different based on whether a sheath becomes stabilized. However, we found that there was no significant difference for this parameter (*Figure 3—figure supplement 2E*). We then divided all stabilized domains by the number of total ensheathment attempts and this results in an a relatively low stabilization rate of 20.5% (*Figure 3—figure supplement 2F*). This means that only one in five ensheathment attempts is stabilized on average by the end of the imaging paradigm at 3 dpf. However, for the instances when the process maintained contact with the axon, 65.1% of these resulted in a stabilized sheath (*Figure 3—figure supplement 2G*). This suggests that continuous axonal contact might promote sheath stabilization. Collectively, this data supports a model where oligodendrocytes extensively explore the axonal environment to find targets that are primed for sheath formation.

## The repetitive ensheathment of axons did not correlate with axon diameter

If the repetitive ensheathment of axons is independent of neuronal activity, it is possible that the mechanism is more biophysical in nature. One possibility is that oligodendrocytes measure the size of each axon by wrapping and unwrapping it multiple times. To explore this possibility, we measured the

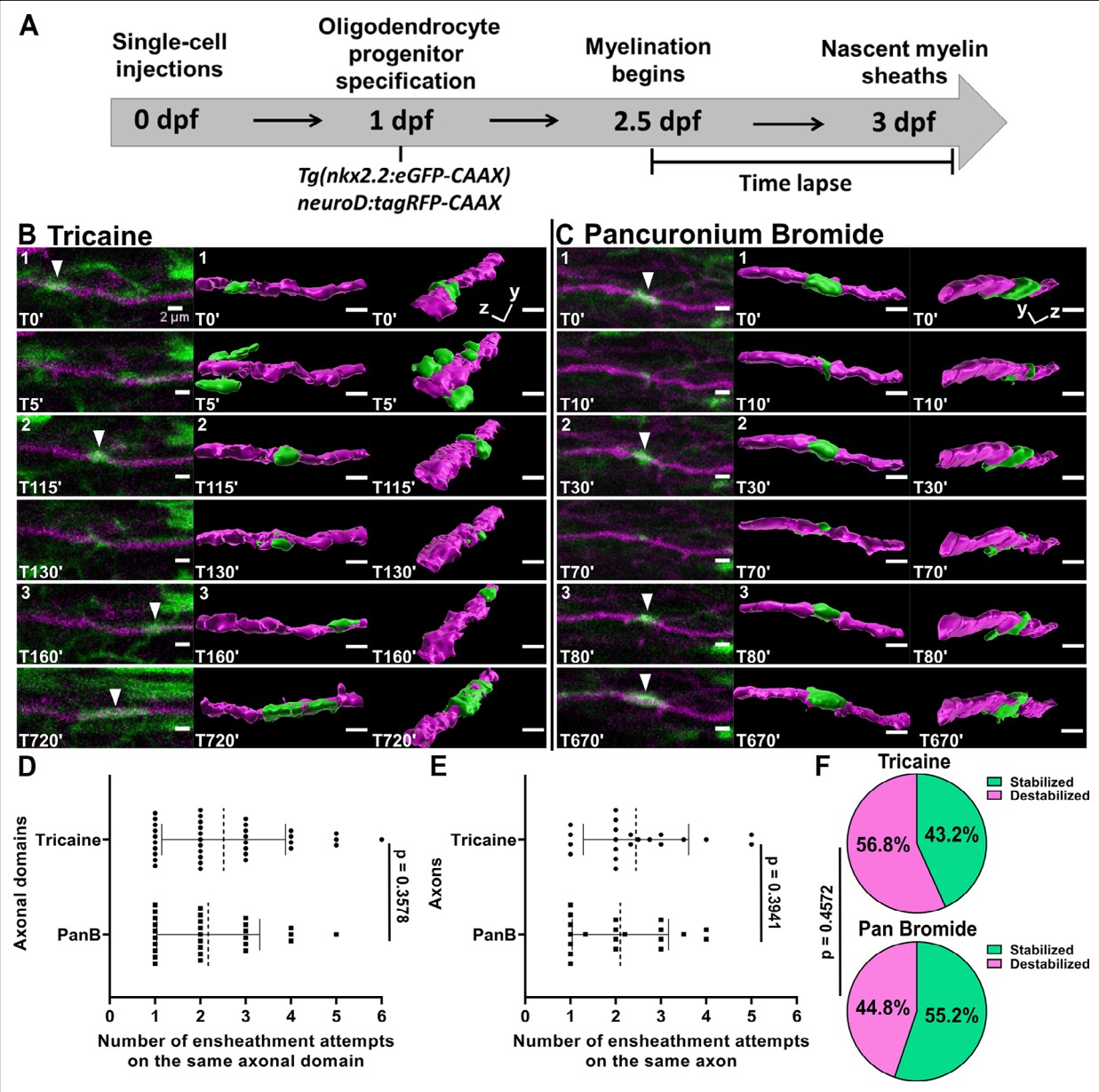

**Figure 3.** Oligodendrocytes repetitively ensheathed axons in an activity-independent manner. (**A**) Axonal ensheathment dynamics experimental paradigm. (**B–C**) Representative lateral images of repetitive ensheathments forming in the spinal cord of living *Tg(nkx2.2:eGFP-CAAX)* (myelin in green) larvae time-lapsed for 18 hr with a 5 min imaging interval starting at 2.5 days post fertilization (dpf). Axons were labeled with *neuroD:tagRFP-CAAX* (magenta). Scale bars = 2 µm. (**B**) Example from larva that was anesthetized in tricaine. (**C**) Example of larva that was anesthetized in pancuronium bromide. (**D**) Plot showing the number of times each axonal domain was ensheathed for the tricaine and pancuronium bromide groups (n=37 axonal domains for tricaine, n=29 axonal domains for pancuronium bromide). (**E**) The same data as in (**D**), but normalized by axon. When more than one domain was observed on an axon, the total number of ensheathments attempts for each domain was averaged (n=21 axons for tricaine collected from 15 larvae, n=20 axons for pancuronium bromide collected from 16 larvae). For (**D**) and (**E**), the dotted lines represent the average number of ensheathment attempts for each group. The error bars represent standard deviation. (**F**) Pie chart of the percent of axonal domains that were stabilized vs. destabilized. The top pie chart is for the tricaine group and the bottom pie chart is for the pancuronium bromide group. Significance was determined using non-parametric Mann-Whitney tests for (**D, E**). Significance was determined using a Fisher's exact test for (**F**) (see associated source data and supplementary video files).

The online version of this article includes the following video, source data, and figure supplement(s) for figure 3:

**Source data 1.** Excel spreadsheet with the repetitive ensheathment numbers and the stabilization status for each axonal domain (tricaine vs pancuronium bromide).

**Figure supplement 1.** Calcium activity in neurons to compare tricaine vs pancuronium bromide forms of anesthesia.

*Figure 3 continued on next page*

*Figure 3 continued*

**Figure supplement 1—source data 1.** Excel spreadsheet with the summary of the calcium transient counts.

**Figure supplement 2.** Summary of oligodendrocyte behavior during the repetitive ensheathment of axons.

**Figure supplement 2—source data 1.** Excel spreadsheet with a summary of the repetitive ensheathment numbers for *Figures 2 and 3*.

**Figure supplement 3.** Axon diameter was not related to the number of ensheathment attempts made on each domain.

**Figure supplement 3—source data 1.** Excel spreadsheet with the axon diameter measurements for the repetitive ensheathment data in *Figures 2 and 3*.

**Figure 3—video 1.** *Figure 3B* fluorescence time-lapse video for the tricaine repetitive ensheathment example.
https://elifesciences.org/articles/82111/figures#fig3video1

**Figure 3—video 2.** *Figure 3B* 3D reconstruction_360° rotation video for the tricaine repetitive ensheathment example.
https://elifesciences.org/articles/82111/figures#fig3video2

**Figure 3—video 3.** *Figure 3C* fluorescence time-lapse video for the pancuronium bromide repetitive ensheathment example.
https://elifesciences.org/articles/82111/figures#fig3video3

**Figure 3—video 4.** *Figure 3C* 3D reconstruction_360° rotation video for the pancuronium bromide repetitive ensheathment example.
https://elifesciences.org/articles/82111/figures#fig3video4

**Figure 3—video 5.** Fluorescence time-lapse video example for the pre-spike control condition.
https://elifesciences.org/articles/82111/figures#fig3video5

**Figure 3—video 6.** Fluorescence time-lapse video example for the corresponding post-spike control condition.
https://elifesciences.org/articles/82111/figures#fig3video6

**Figure 3—video 7.** Fluorescence time-lapse video example for the pre-tricaine spike condition.
https://elifesciences.org/articles/82111/figures#fig3video7

**Figure 3—video 8.** Fluorescence time-lapse video example for the corresponding post-tricaine spike condition.
https://elifesciences.org/articles/82111/figures#fig3video8

diameters of all the axonal domains in our two repetitive ensheathment experiments from *Figures 2 and 3* above. We performed a full-width 1/3 max measurement of the diameter of each axonal domain underneath one of the ensheathments in each series. We then did a linear regression to assess the relationship between the diameter of the axon and the number of times that it was ensheathed. The $R^2$ value was 0.01 regardless of whether the ensheathment was stabilized or not. Thus, we found no relationship between these parameters in our data (*Figure 3—figure supplement 3A–C*).

## Oligodendrocyte nascent sheath accumulation involved extensive sheath initiation and loss

In the previous axonal ensheathment experiments, we captured the dynamics of sheath initiation from the perspective of the individual processes of an oligodendrocyte. We next sought to understand how these dynamics contribute to the overall sheath number of entire oligodendrocytes. To do this, we modified the ensheathment dynamics imaging paradigm from *Figures 2 and 3* (*Figure 4A*). Embryos were co-injected with our *sox10:eGFP-CAAX* oligodendrocyte lineage cell membrane reporter and *myrf:tagRFP*, a cytosolic reporter expressed in myelin-fated oligodendrocytes. This allowed us to identify sparse, double-labeled eGFP+RFP+ cells in both the dorsal and ventral tracts of the spinal cord at 2.5 dpf. We captured the cellular dynamics of each cell for 15 hr with a 5 min imaging interval. At 3 dpf, larvae were placed back into embryo medium. At 4 dpf, the same larvae were remounted, and a single static image was taken for each of the same cells. From this, the formation and loss of every ensheathment could be tracked for each cell. Additionally, quality control experiments determined that the conditions of this imaging paradigm did not change the final sheath number or average sheath length of these cells (*Figure 4—figure supplement 1*, see Materials and methods for further details).

To begin answering the question of how sheath initiation and loss is balanced for individual oligodendrocytes, every ensheathment was manually tracked and quantified in each frame of our time-lapse data set. As in published studies (*Czopka et al., 2013*), we found that oligodendrocytes have an initial phase of immature sheath accumulation (*Figure 4B*), followed by a phase of sheath stabilization and loss (*Figure 4C* and *Figure 4—figure supplement 2*). However, the accumulation phase was very dynamic in our studies, with a significant number of ensheathments eventually lost. For example, only

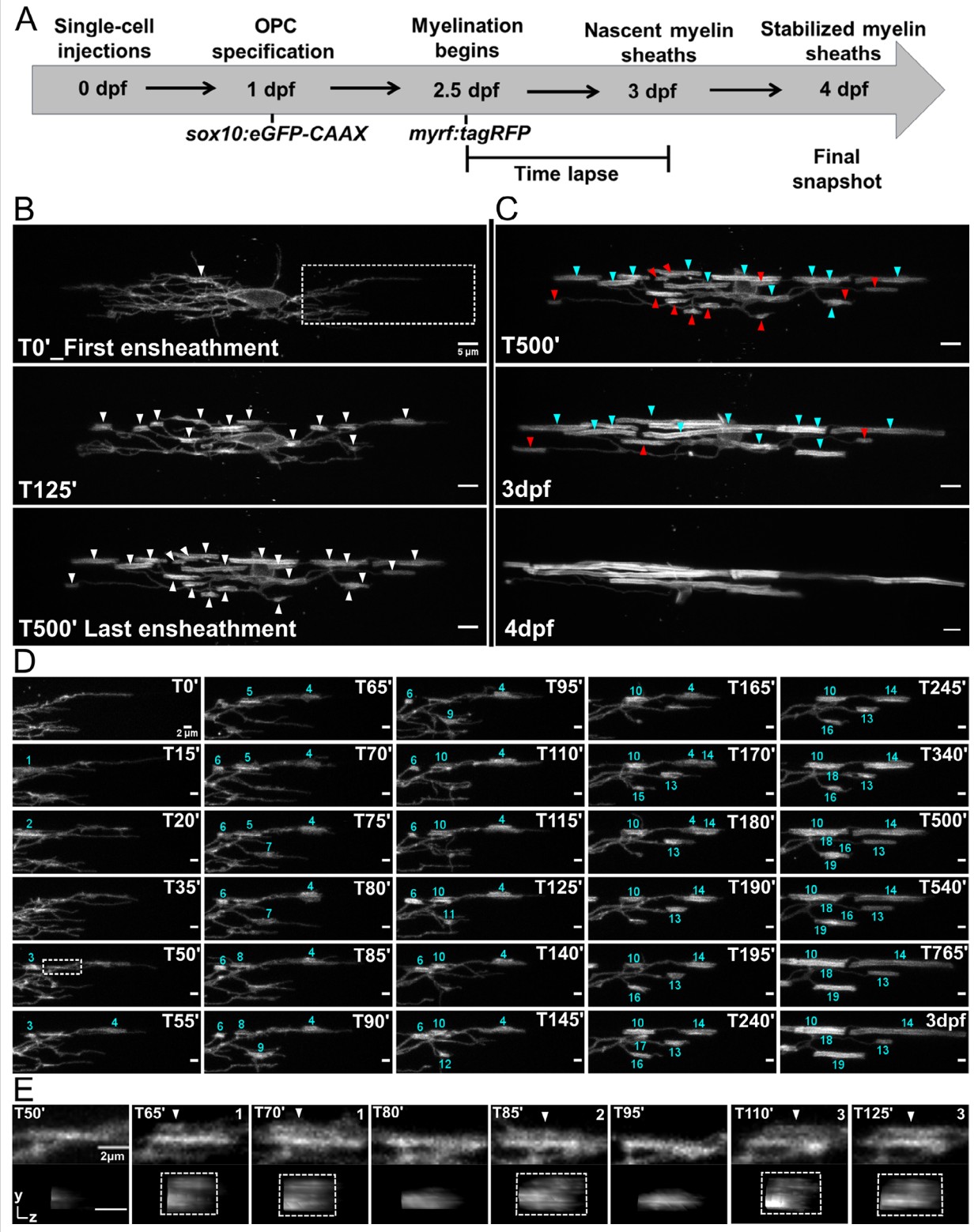

**Figure 4.** Oligodendrocyte nascent sheath accumulation involved extensive sheath initiation and loss. (**A**) Oligodendrocyte ensheathment dynamics imaging paradigm. (**B–E**) Lateral images of oligodendrocyte ensheathment dynamics in the spinal cord of living larvae labeled with *sox10:eGFP-CAAX* and time-lapsed for 15 hours with a 5-minute imaging interval from 2.5-3dpf. (**B**) Images show the progression of immature sheath accumulation from the first ensheathment attempt in the first panel (set at T0') to the final ensheathment at T500' in the bottom panel. Outlined region in first panel is analyzed further in Figure D and E. White arrows identify all immature ensheathments in each frame. (Scale bar = 5 μm). (**C**) Images showing sheath loss

*Figure 4 continued on next page*

*Figure 4 continued*

during the stabilization phase. T500' image is presented again in the top panel and is relabeled to identify future stabilized (cyan) or destabilized (red) ensheathments. This oligodendrocyte has 22 immature sheaths at T500', 11 of which disappeared, and it has 11 stabilized sheaths at 4dpf in the final panel (*Figure 4—figure supplement 2*). (Scale bar = 5 μm). (**D**) Images of the outlined region in the first panel of B visualizing frequent sheath initiation and loss. Each ensheathment is represented with the numbers 1-19 to signify the order that each one appears. The numbers disappear when an ensheathment is lost. Only 5 ensheathment attempts were stabilized out of a total of 19. (Scale bar = 2 μm) (**E**) Further enlarged images of the outlined region in Figure D (T50'). White arrows point out 3 ensheathment attempts before the final one was stabilized. The YZ images are cross sections of each ensheathment (volume projections in Imaris). A box is drawn around each cylindrical ensheathment in the YZ images. (Scale bar = 2 μm). (See associated source data and supplementary video files).

The online version of this article includes the following video, source data, and figure supplement(s) for figure 4:

**Figure supplement 1.** Ensheathment dynamics imaging conditions did not significantly alter sheath number or length.

**Figure supplement 1—source data 1.** Excel spreadsheet with the sheath analysis data comparing the standard condition, tricaine and agar condition, and the time-lapse condition.

**Figure supplement 2.** Visual quantification of sheath number at 4 days post fertilization (dpf) for the representative cell in *Figure 4C*.

**Figure 4—video 1.** *Figure 4B and C* fluorescence time-lapse video for the oligodendrocyte ensheathment dynamics example.

https://elifesciences.org/articles/82111/figures#fig4video1

**Figure 4—video 2.** *Figure 4D* fluorescence time-lapse video for inset panel in 4B.

https://elifesciences.org/articles/82111/figures#fig4video2

5 out of 19 ensheathment attempts were stabilized for the outlined region of the cell in *Figure 4B* (# 10, 13, 14, 18, and 19, *Figure 4D*). This is consistent with our data in *Figure 3—figure supplement 2F*. We also readily observed instances of possible repetitive ensheathment (*Figure 4E*). However, since we do not have axons labeled in this experiment, we cannot definitively conclude that this is what we observed.

To better understand the impact of these ensheathment dynamics, we needed to average the levels of sheath initiation and loss across all cells in our data set and establish a graphic method for presenting the results. However, the number of hours over which each cell accumulates sheaths ranged from ~4 to 8 hr (*Figure 5A*), making it difficult to combine data from multiple cells for quantitative comparison. Nevertheless, we identified a definable transition point for each cell from when they were accumulating new immature sheaths to when they were stabilizing them. Thus, once an oligodendrocyte accumulated a peak (or maximum) number of immature sheaths, every cell slowed down and stopped forming new sheaths. Given this common dynamic of accumulation and plateau, it was therefore possible to normalize the timing of sheath accumulation in each video by defining the frame at which each cell had accumulated its peak number of sheaths as time zero for that cell (*Figure 5B*). Using this normalization for 19 individual dorsal cells and 18 individual ventral cells, we found that the dorsal cells accumulated a higher peak number of ensheathments compared to the ventral cells (*Figure 5B*). Additionally, ~95% of all ensheathments attempts occurred before reaching this peak for both groups (*Figure 5C* and *Figure 5D*, vertical dashed line). This was an average of ~100 ensheathment attempts for dorsal cells and ~74 for ventral cells. Interestingly, ~75% of these ensheathments were also lost before reaching peak for both groups (*Figure 5E and F*). Collectively, this data shows that oligodendrocytes initiate a remarkable number of immature ensheathments, with only a small percentage maintained by the end of the accumulation phase.

## All oligodendrocytes exhibited a low and consistent sheath stabilization rate

It is very striking that all oligodendrocytes exhibited a similar rate of sheath loss during the accumulation phase (*Figure 5*), yet dorsal cells end up with more sheaths than ventral cells. Dorsal oligodendrocytes exhibited more ensheathment attempts, accumulated a higher peak number of immature ensheathments, and maintained more sheaths at the 4 dpf time point compared to ventral cells (*Figure 6A–E*). Dorsal oligodendrocytes also lost more sheaths during the stabilization phase compared to ventral cells (*Figure 6F*). However, both populations stabilized a similar percentage of ensheathments throughout the accumulation, stabilization, and combined phases of our experimental paradigm (*Figure 6G–I*). Thus, the levels of sheath initiation for each cell are very heterogeneous (ranging from ~20 to 190), but all oligodendrocytes exhibit a similar overall rate of sheath stabilization

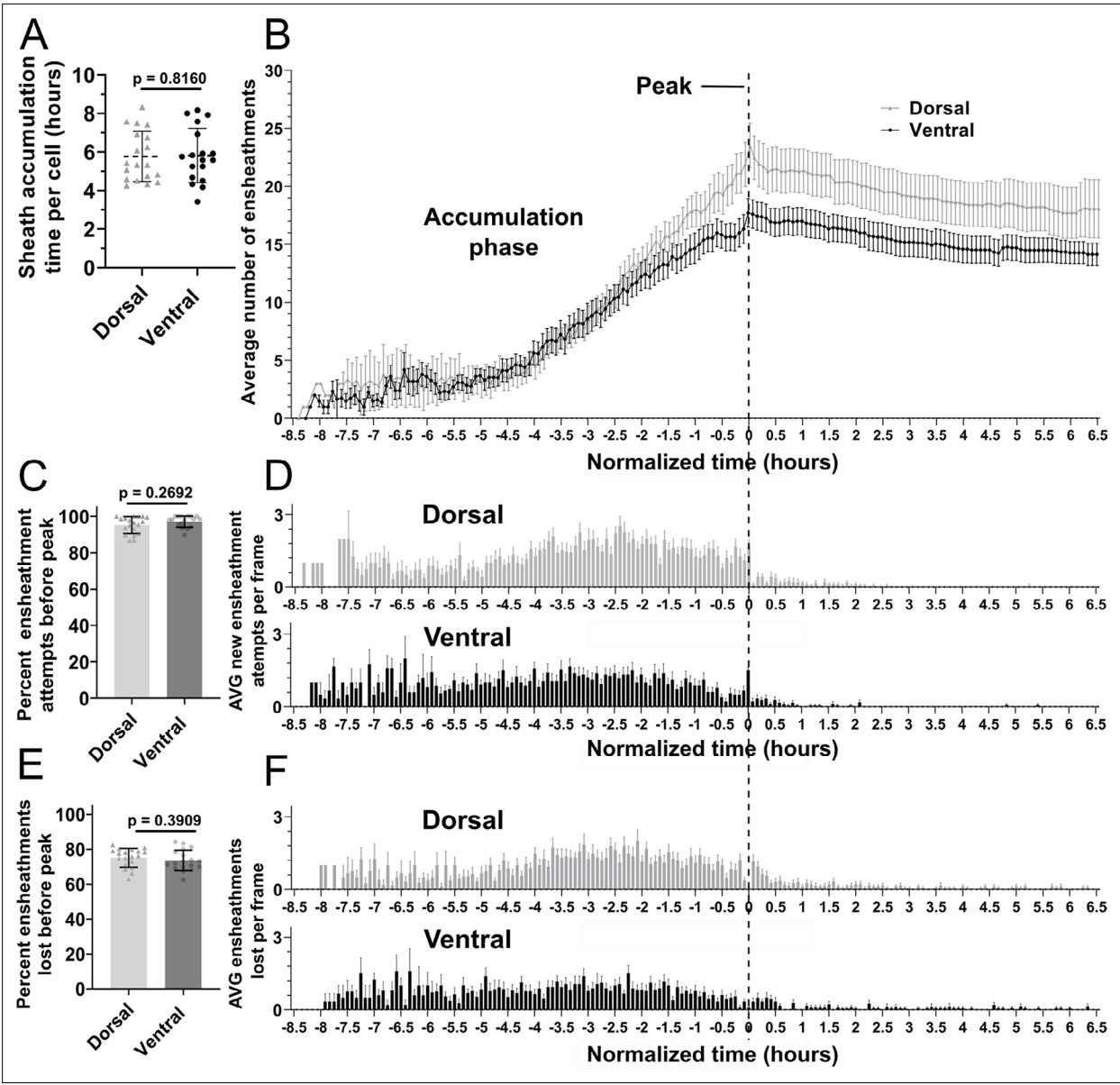

**Figure 5.** Oligodendrocyte nascent sheath accumulation involved a consistent and high rate of sheath initiation and loss. (**A**) Time required for peak (or max) sheath accumulation per cell (hours). Error bars are standard deviation. (**B–F**) Videos of n=19 dorsal cells (19 larvae) and n=18 ventral cells (18 larvae) were quantified for sheath initiation and loss in each 5 min frame. The time-lapse videos were then normalized for quantitative comparison by defining the video frame at which each cell accumulated its peak (or max) number of immature ensheathments as T0'. Each frame before or after that was –5 min or +5 min. The vertical dashed line aligns the T0' time point of each graph in B, D, and F. (**B**) The average number of immature ensheathments for dorsal and ventral cells that are present in each frame is plotted based on time relative to T0'; error bars represent SEM. (**C**) Percent of ensheathment attempts from the entire imaging period that occurred prior to reaching peak. Error bars represent SEM. (**D**) The average number of new ensheathment attempts in each frame is plotted with the same time normalization as in B. Error bars represent SEM. (**E**) Similar to C, the percent of ensheathments from the entire imaging period that were lost prior to reaching peak. Error bars represent SEM. (**F**) The average number of ensheathments that were lost in each frame is plotted with the same time normalization as in B. Error bars represent SEM. Individual data points are shown and significance was determined by Mann-Whitney tests in A, C, and E. The data in B, D, and F was cropped at +6.5 hr (see associated source data).

The online version of this article includes the following source data for figure 5:

**Source data 1.** Excel spreadsheet with the sheath accumulation data for the dorsal and ventral oligodendrocyte ensheathment dynamics experiment.

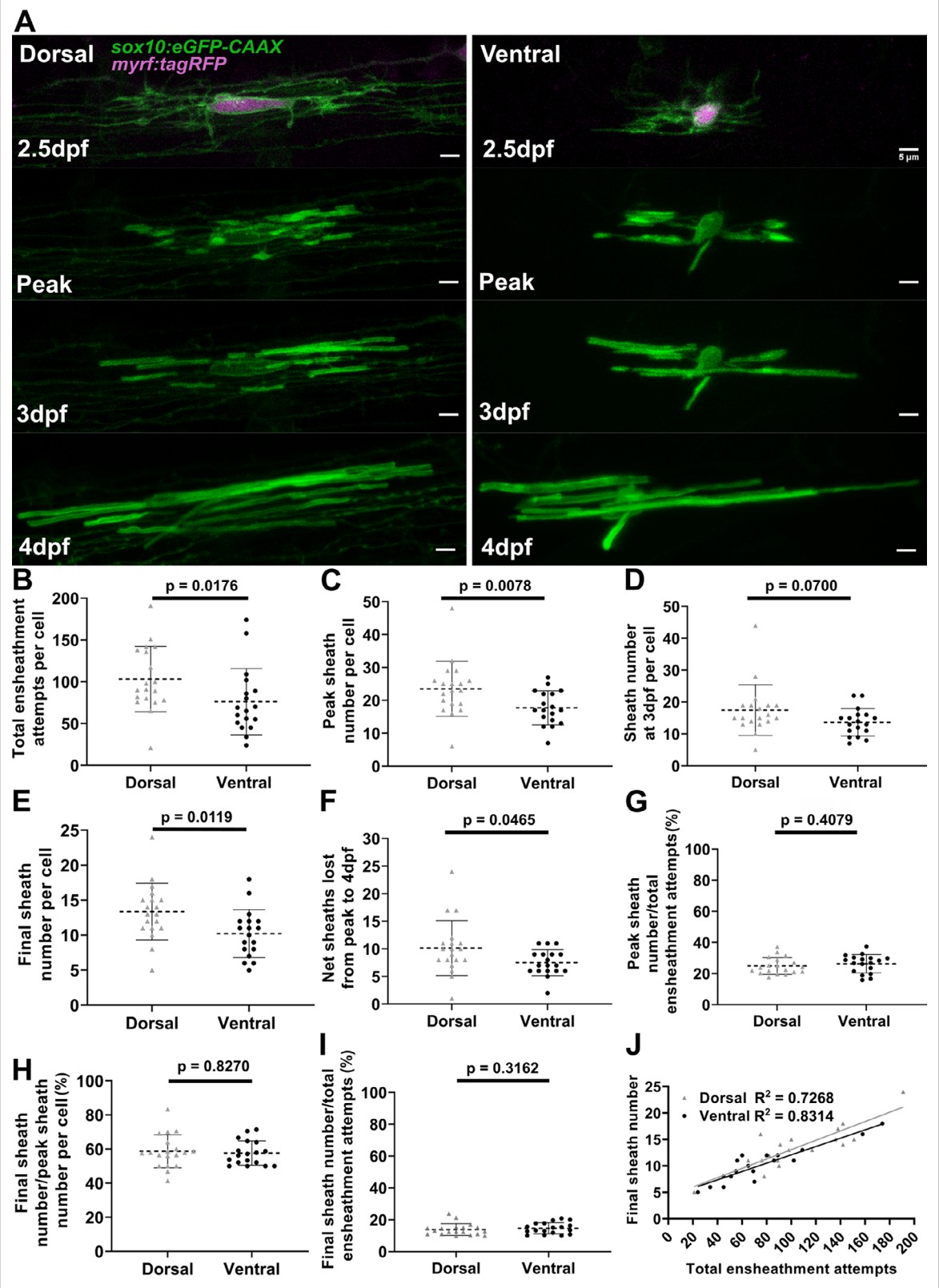

**Figure 6.** Dorsal cells initiated more ensheathments than ventral cells, but all oligodendrocytes stabilized a similar percentage of these ensheathments. (**A**) Dorsal and ventral cell lateral images from the ensheathment dynamics imaging paradigm in the spinal cord of living larvae labeled with *sox10:eGFP-CAAX* (in green) and time-lapse imaged for 15 hr from 2.5 to 3 days post fertilization (dpf). The upper panels are a dorsal (left) and ventral (right) cell also labeled with *myrf:tagRFP* (magenta) at the beginning of the time-lapse experiment. The subsequent panels are the same cells at the

*Figure 6 continued on next page*

*Figure 6 continued*

peak of sheath accumulation, at 3 and 4 dpf (scale bar = 5 µm). (**B–J**) compares dorsal and ventral cells. (**B**) Total ensheathment attempts per cell. (**C**) Peak sheath number per cell. (**D**) Sheath number at 3 dpf per cell. (**E**) Final sheath number per cell at 4 dpf. (This is the same data and images as presented in *Figure 4—figure supplement 1*, time-lapse group.) (**F**) Net sheaths lost from the peak to 4 dpf. (**G**) Percent of sheaths stabilized during the accumulation phase (peak sheath number/total ensheathment attempts). (**H**) Percent of sheaths stabilized during the stabilization phase (final sheath number/peak sheath number). (**I**) Percent of total sheaths stabilized across both the accumulation and stabilization phases (final sheath number/total ensheathment attempts). (**J**) Simple linear regression comparing the total number of ensheathment attempts to the final sheath number at 4 dpf for each cell. The $R^2$ values for each group are shown. Dorsal n=19 cells/19 larvae, ventral n=18 cells/18 larvae. The dashed lines in each plot represent average values with all data points shown. The error bars are standard deviation. Significance was determined by Mann-Whitney tests (see associated source data and supplementary video files).

The online version of this article includes the following video, source data, and figure supplement(s) for figure 6:

**Source data 1.** Excel spreadsheet with the summary data for the dorsal and ventral oligodendrocyte ensheathment dynamics experiment.

**Figure supplement 1.** Oligodendrocytes did not make new sheaths during the stabilization phase from 3 to 4 days post fertilization (dpf).

**Figure supplement 1—source data 1.** Excel spreadsheet with the summary data for the oligodendrocyte ensheathment dynamics experiment looking at 3–4 days post fertilization (dpf).

**Figure 6—video 1.** *Figure 6A* fluorescence time-lapse video for the dorsal oligodendrocyte ensheathment dynamics example.
https://elifesciences.org/articles/82111/figures#fig6video1

**Figure 6—video 2.** *Figure 6A* fluorescence time-lapse video for the ventral oligodendrocyte ensheathment dynamics example.
https://elifesciences.org/articles/82111/figures#fig6video2

**Figure 6—video 3.** *Figure 6—figure supplement 1* fluorescence time-lapse video for the 2.5–3 days post fertilization (dpf) oligodendrocyte ensheathment dynamics example.
https://elifesciences.org/articles/82111/figures#fig6video3

**Figure 6—video 4.** *Figure 6—figure supplement 1* fluorescence time-lapse video for the corresponding 3–4 days post fertilization (dpf) oligodendrocyte ensheathment dynamics example.
https://elifesciences.org/articles/82111/figures#fig6video4

(~10–20%). To be confident with our quantification of this stabilization rate, we needed to be sure that these cells did not produce any new sheaths from 3 to 4 dpf during the sheath stabilization phase. We designed a modified ensheathment dynamics imaging paradigm to test this and found that oligodendrocytes made no new sheaths during this time interval (*Figure 6—figure supplement 1*, see Materials and methods for further details). Collectively, we conclude that dorsal cells end up with more sheaths at 4 dpf compared to ventral cells due to increased sheath initiation.

We predict that this consistent sheath stabilization rate, irrespective of sheath initiation, could be a fundamentally important parameter for understanding oligodendrocyte ensheathment behavior. Indeed, a simple linear regression comparing the total number of ensheathment attempts with the final number of sheaths for each cell established high correlation of these parameters ($R^2$ value of ~0.73 for dorsal cells and ~0.83 for ventral cells, *Figure 6J*). Thus, for each stabilized ensheathment, there is a relatively predictable number of ensheathments that are lost. Altogether, these results strongly suggest that while both sheath initiation and loss vary among oligodendrocytes, they are proportionately regulated across all oligodendrocytes.

## Components of the endocytic recycling pathway localized to immature sheaths and regulated myelin sheath number

The repetitive ensheathment of axons is orchestrated by a dynamic series of morphological transitions that are energetically expensive for oligodendrocytes. Thus, it seems plausible that this process is fundamental for the proper accumulation and stabilization of sheaths. The endocytic recycling pathway is responsible for the reuse or turnover of membrane and membrane proteins in a cell (depicted in *Figure 7A*). We therefore decided to determine the importance of these early sheath initiation dynamics by disrupting the endocytic recycling pathway.

The endocytic recycling pathway is controlled by several Rab-GTPase proteins that act as master regulators of membrane trafficking (reviewed in *Pfeffer, 2017*). Rab5, Rab7, and Rab11 regulate the early endosome, late endosome/lysosome, and recycling endosome, respectively (*Figure 7A*). The cargo of endocytic vesicles is sorted at the early endosome (Rab5) and is then trafficked back to the plasma membrane or to the recycling endosome (Rab11) for further sorting. Alternatively, molecular

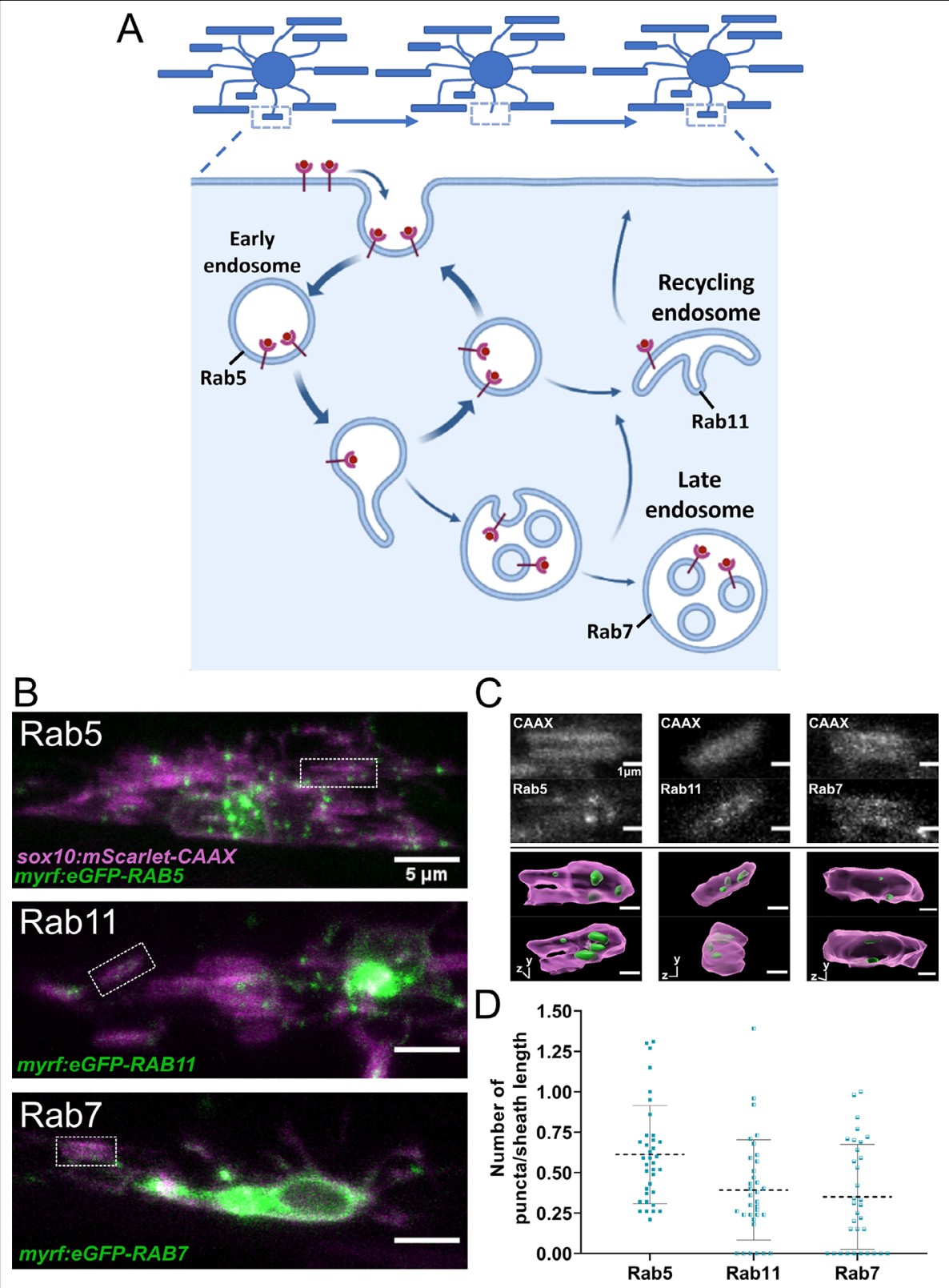

**Figure 7.** Rab5+, Rab7+, and Rab11+ endosomes localized to immature sheaths. (**A**) The endocytic recycling pathway during sheath initiation and loss. (**B**) Lateral images of oligodendrocytes in the early stages of the ensheathment process in the spinal cord of living larvae at 2.5 dpf labeled with *sox10:mScarlet-CAAX* (magenta) and expressing either *myrf:eGFP-RAB5C, myrf:eGFP-RAB7A,* or *myrf:eGFP-RAB11A* (green). White boxes outline immature sheaths with Rab+ endosomal puncta for each fusion protein. (Scale bar = 5 μm). (**C**) Top panels are grey inset images from the outlined

*Figure 7 continued on next page*

*Figure 7 continued*

regions in B. The bottom panels are 3D reconstructions of these insets (Membrane in magenta, endosomes in green). (**D**) Quantification of Rab+ endosomal puncta in immature sheaths. Number of puncta in each sheath was normalized by the length of the sheath. Rab5 n = 36 sheaths (9 ventral cells/2 dorsal cells/11 larvae), Rab11 n = 34 sheaths (8 ventral cells/2 dorsal cells/10 larvae), Rab7 n = 33 sheaths (9 ventral cells/2 dorsal cells/11 larvae). Dashed lines represent average values and error bars are SD. (See associated source data and supplementary video files).

The online version of this article includes the following video and source data for figure 7:

**Source data 1.** Excel spreadsheet with the Rab5, -7, -11 localization data.

**Figure 7—video 1.** *Figure 7C* 3D reconstruction_360° rotation video for the Rab5 inset example.
https://elifesciences.org/articles/82111/figures#fig7video1

**Figure 7—video 2.** *Figure 7C* 3D reconstruction_360° rotation video for the Rab11 inset example.
https://elifesciences.org/articles/82111/figures#fig7video2

**Figure 7—video 3.** *Figure 7C* 3D reconstruction_360° rotation video for the Rab7 inset example.
https://elifesciences.org/articles/82111/figures#fig7video3

cargo will be degraded as early endosomes mature into late endosomes/lysosomes (Rab7) (*Grant and Donaldson, 2009*; *Langemeyer et al., 2018*).

We initially analyzed whether the Rab5, Rab7, or Rab11 proteins localized to immature myelin sheaths. Expression constructs were generated to specifically label Rabs in oligodendrocytes, using *myrf* regulatory DNA to drive eGFP-Rab fusion proteins: *myrf:eGFP-RAB5C*, *myrf:eGFP-RAB7A*, and *myrf:eGFP-RAB11A*. We transiently expressed each Rab fusion protein with *sox10:mScarlet-CAAX* to label oligodendrocyte membranes. Images of early myelinating oligodendrocytes expressing these Rab proteins were collected at 2.5 dpf (*Figure 7B and C*), and the number of Rab+ puncta present in each immature ensheathment was counted and normalized by the length of each sheath. All three types of endosomes were present within immature ensheathments (*Figure 7D*). This data suggests that Rab5, Rab7, and Rab11 could all potentially play a role in sheath formation.

To investigate whether Rab5, Rab7, or Rab11 are important for regulating sheath number in oligo-dendrocytes, we performed a sheath analysis in cells expressing dominant-negative Rab mutants (*Clark et al., 2011*; *Zhang et al., 2007*). Rab proteins are GTPases that are active when bound to GTP and inactive when bound to GDP (*Müller and Goody, 2018*). Thus, the $rab5C^{S36N}$, $rab7A^{T22N}$, and $rab11A^{S25N}$ dominant-negative point mutations maintain the Rab protein in a GDP-bound 'off' state. However, since these mutant proteins still bind GEF proteins that normally activate Rabs, they reduce the level of GEFs available to activate endogenous Rabs, thereby decreasing the overall activity of the endogenous Rab protein.

We injected $myrf:tagRFP-rab5C^{S36N}$, $myrf:tagRFP-rab7A^{T22N}$, and $myrf:tagRFP-rab11A^{S25N}$ fusion constructs into embryos alongside *sox10:eGFP-CAAX* to label oligodendrocyte membranes. We captured static images of ventral cells only at 4 dpf and quantified myelin sheath number and sheath length. We found that the Rab5DN and Rab11DN mutants both decreased the average sheath number per cell without changing average sheath length (*Figure 8A–D*). Thus, expression of these mutants apparently had no general negative impact on oligodendrocyte development, but they likely regulate a specific aspect of the ensheathment process. On the other hand, the Rab7DN mutant had little impact on sheath number or sheath length (*Figure 8A–D*), which suggests that the phenotype we observe for the Rab5DN and Rab11DN mutants is specific to the recycling pathway and is not a generic effect of expressing a dominant-negative Rab mutant.

## Rab5 regulated longitudinal sheath stability

We hypothesize that the repetitive ensheathment of axons is required to facilitate optimal sheath accumulation and stabilization and is regulated by the endocytic recycling pathway. We therefore predict that disrupting endocytic recycling will reduce these dynamics and diminish overall sheath accumulation. Both Rab5 and Rab11 are involved in the recycling component of the endocytic pathway and their dominant-negative mutants both decreased the overall sheath number per cell. Therefore, we decided to specifically investigate Rab5 in depth. We analyzed cells expressing the Rab5DN mutant

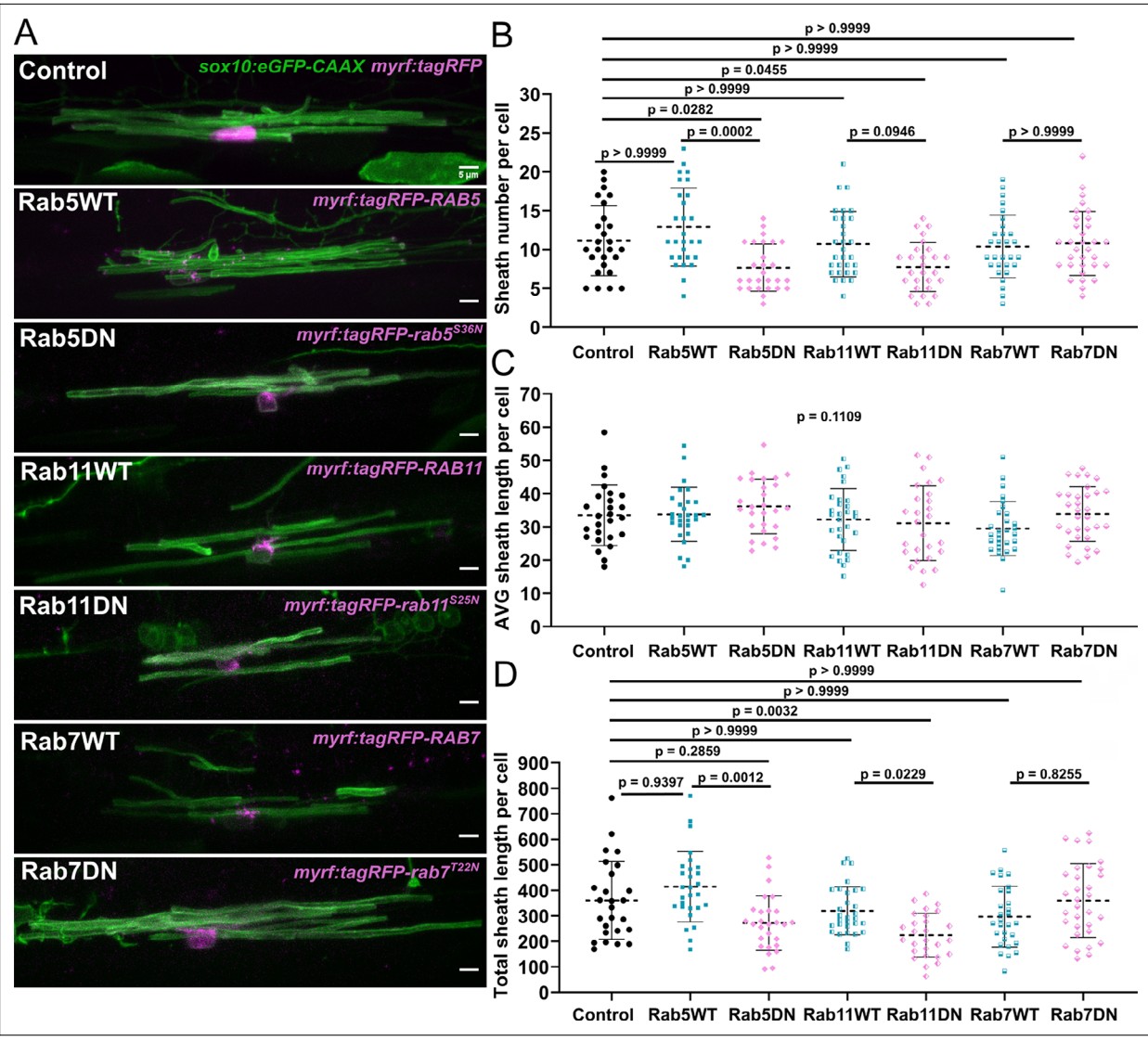

**Figure 8.** The Rab5 and Rab11 dominant-negative mutants reduced sheath number, but not average sheath length. (**A**) Representative lateral images of ventral oligodendrocytes in the spinal cord of living larvae at 4 days post fertilization (dpf) labeled by *sox10:eGFP-CAAX* (green) and one of the following: *myrf:tagRFP, myrf:tagRFP-RAB5C, myrf:tagRFP-rab5C^S36N, myrf:tagRFP-RAB7A, myrf:tagRFP-rab7A^T22N,* and *myrf:tagRFP-RAB11A, myrf:tagRFP-rab11A^S25N* (all in magenta). The image and data for the control is the same as for the ventral group in *Figure 1* (scale bar = 5 µm). (**B**) Sheath number per cell. (**C**) Average sheath length per cell. (**D**) Total sheath length per cell (*myrf:tagRFP* n=26 cells/26 larvae, *myrf:tagRFP-RAB5C* n=28 cells/28 larvae, *myrf:tagRFP-rab5C^S36N* n=27 cells/27 larvae, *myrf:tagRFP-RAB7A* n=29 cells/29 larvae, *myrf:tagRFP-rab7A^T22N* n=32 cells/32 larvae, and *myrf:tagRFP-RAB11A* n=30 cells/30 larvae, *myrf:tagRFP-rab11A^S25N* n=27 cells/27 larvae). The dashed lines in each plot represent average values with all data points shown. Error bars are standard deviation. Global significance was determined using a Kruskal-Wallis test for B–D. This global p-value is shown for C since it was not significant. Post hoc multiple comparison tests were not performed for this analysis. Post hoc Dunn's multiple comparison tests were performed to compare groups in B and D. We compared everything with the control group and compared each wild-type and associated mutant with each other. The different Rab groups were not compared with each other (see associated source data).

The online version of this article includes the following source data for figure 8:

**Source data 1.** Excel spreadsheet with the sheath analysis data for the Rab5, -7, -11 dominant-negative mutants.

in the ventral spinal cord tracts using the same 'oligodendrocyte ensheathment dynamics paradigm' from *Figures 4–6* above. Oligodendrocytes expressing either *myrf:tagRFP-RAB5C* or *myrf:tagRFP-rab5C^S36N* were imaged and compared to the ventral control data from *Figures 4–6*.

Unexpectedly, the over-expression of the Rab5DN mutant did not change the dynamics of the accumulation phase. These cells formed and lost the same number of ensheathments during this phase and accumulated ensheathments to a similar peak number compared to the controls (*Figure 9A–C*,

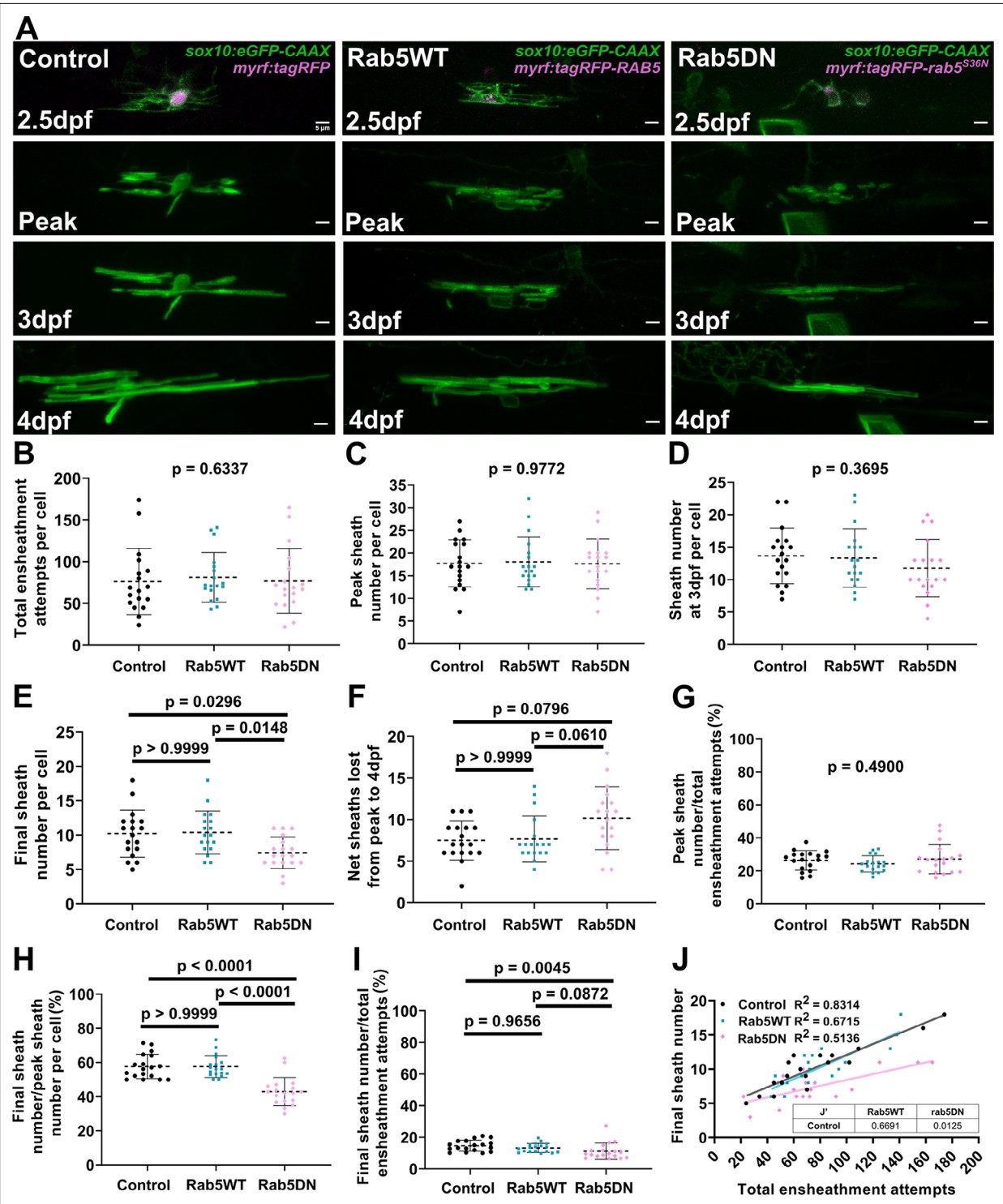

**Figure 9.** Over-expression of the Rab5 dominant-negative mutant in oligodendrocytes reduced sheath stability in the stabilization phase. (**A**) Lateral images from the ventral spinal cord of living larvae labeled with *sox10:eGFP-CAAX* (in green) and one of the following: *myrf:tagRFP*, *myrf:tagRFP-RAB5C*, *myrf:tagRFP-rab5C^{S36N}* (all in magenta); and time-lapsed for 15 hours from 2.5-3dpf. The first panel is an image taken immediately before starting the time-lapse experiment. The subsequent panels are the same cells at the peak of sheath accumulation, at 3dpf, and at 4dpf. The images and data for the control are the same as for the ventral group in *Figure 6*. (Scale bar = 5 µm). (**B**) Total ensheathment attempts per cell. (**C**) Peak sheath number per cell. (**D**) Sheath number at 3dpf per cell. (**E**) Final sheath number per cell at 4dpf. (**F**) Net sheaths lost from the peak to 4dpf. (**G**) Percent of sheaths stabilized during the accumulation phase (peak sheath number/total ensheathment attempts). (**H**) Percent of sheaths stabilized during the stabilization phase (final sheath number/peak sheath number). (**I**) Percent of total sheaths stabilized across both the accumulation and stabilization phases (final

*Figure 9 continued on next page*

*Figure 9 continued*

sheath number/total ensheathment attempts). (**J**) Simple linear regression comparing the total number of ensheathment attempts to the final sheath number at 4dpf for each cell. (control n=18 cells/18 larvae, wild-type Rab5 n=18 cells/18 larvae, Rab5DN n=18 cells/17 larvae). The dashed lines in each plot represent average values with all data points shown. The error bars are standard deviation. Significance was determined using global Kruskal-Wallis tests. These p-values are shown for B-D and G since they were not significant. Post hoc multiple comparisons tests were not performed for these analyses. Post hoc Dunn's multiple comparisons tests were done for E, F, H, and I and the individual p-values are shown. (**J′**) The slopes of the Rab5WT and Rab5DN regression lines from J were compared to the control in Graphpad by (two-tailed) testing the null hypothesis that the slopes are identical (the lines are parallel). P-values are shown in the table. (See associated source data and supplementary video files).

The online version of this article includes the following video, source data, and figure supplement(s) for figure 9:

**Source data 1.** Excel spreadsheet with the summary data for the Rab5 dominant-negative mutant oligodendrocyte ensheathment dynamics experiment.

**Figure supplement 1.** The dynamics of the sheath accumulation phase were not altered by Rab5DN over-expression.

**Figure supplement 1—source data 1.** Excel spreadsheet with the sheath accumulation data for the Rab5 dominant-negative mutant oligodendrocyte ensheathment dynamics experiment.

**Figure 9—video 1.** Fluorescence time-lapse video for the control ventral oligodendrocyte ensheathment dynamics example.
https://elifesciences.org/articles/82111/figures#fig9video1

**Figure 9—video 2.** Fluorescence time-lapse video for the Rab5WT ventral oligodendrocyte ensheathment dynamics example.
https://elifesciences.org/articles/82111/figures#fig9video2

**Figure 9—video 3.** Fluorescence time-lapse video for the Rab5DN ventral oligodendrocyte ensheathment dynamics example.
https://elifesciences.org/articles/82111/figures#fig9video3

*Figure 9—figure supplement 1A–F*). Instead, the Rab5DN mutant specifically impacted the stabilization phase (which starts after cells reach peak sheath accumulation). We found a strong trend that the mutant increased the net number of sheaths that were lost from the peak sheath number out to 4 dpf although it was not statistically significant (*Figure 9C–F*). Consistent with this trend, cells over-expressing the Rab5DN mutant stabilized a lower percentage of sheaths in the stabilization phase specifically (*Figure 9G–I*). A simple linear regression comparing the total ensheathment attempts to the final sheath number showed that the slope of the mutant regression line was significantly lower than for the control group (*Figure 9J*). Collectively, we conclude that the dominant-negative Rab5 mutant expressed from the *myrf* driver negatively impacts the longitudinal sheath stabilization rate of oligodendrocytes but has little or no impact on initial sheath accumulation.

## Discussion

In this study we examined how sheath initiation and loss are balanced to regulate sheath number in oligodendrocytes. We combined extensive time-lapse and longitudinal imaging of single oligodendrocytes and demonstrated that it was the total number of initial ensheathments formed that determined sheath number. Our analyses revealed that this was because ~80–90% of all ensheathments were consistently lost by each cell. This stabilization rate was comparable, whether we imaged the behavior of individual oligodendrocytes (*Figures 4–6*) or of individual axons (*Figures 2 and 3*). After summing up all the ensheathments that were formed in the axonal experiments, ~80% of them were lost. We therefore concluded that this low and consistent stabilization rate resulted from the repetitive ensheathment of axons (*Figure 10A*), which was independent of neuronal activity in our experiments. We next tested whether these dynamics were required for the proper accumulation of sheaths during the sheath initiation phase of myelination. It is likely that the same membrane and membrane proteins are being used with each iterative ensheathment attempt and so we disrupted the endocytic recycling pathway to perturb these dynamics. We found that inhibiting the activity of Rab5, a GTPase protein that regulates trafficking at the early endosome, caused oligodendrocytes to lose more sheaths during the stabilization phase specifically (*Figure 10B*). We therefore can only definitively conclude that Rab5 and endocytic recycling regulates longitudinal sheath stabilization in oligodendrocytes, i.e., its role in sheath initiation is unclear from these experiments.

The extensive loss of ensheathments that we observed in our experiments seems to contrast with previous studies reporting that oligodendrocytes maintain most of the sheaths that are formed (*Czopka et al., 2013*; *Mensch et al., 2015*). Those studies were the first to demonstrate that oligodendrocytes

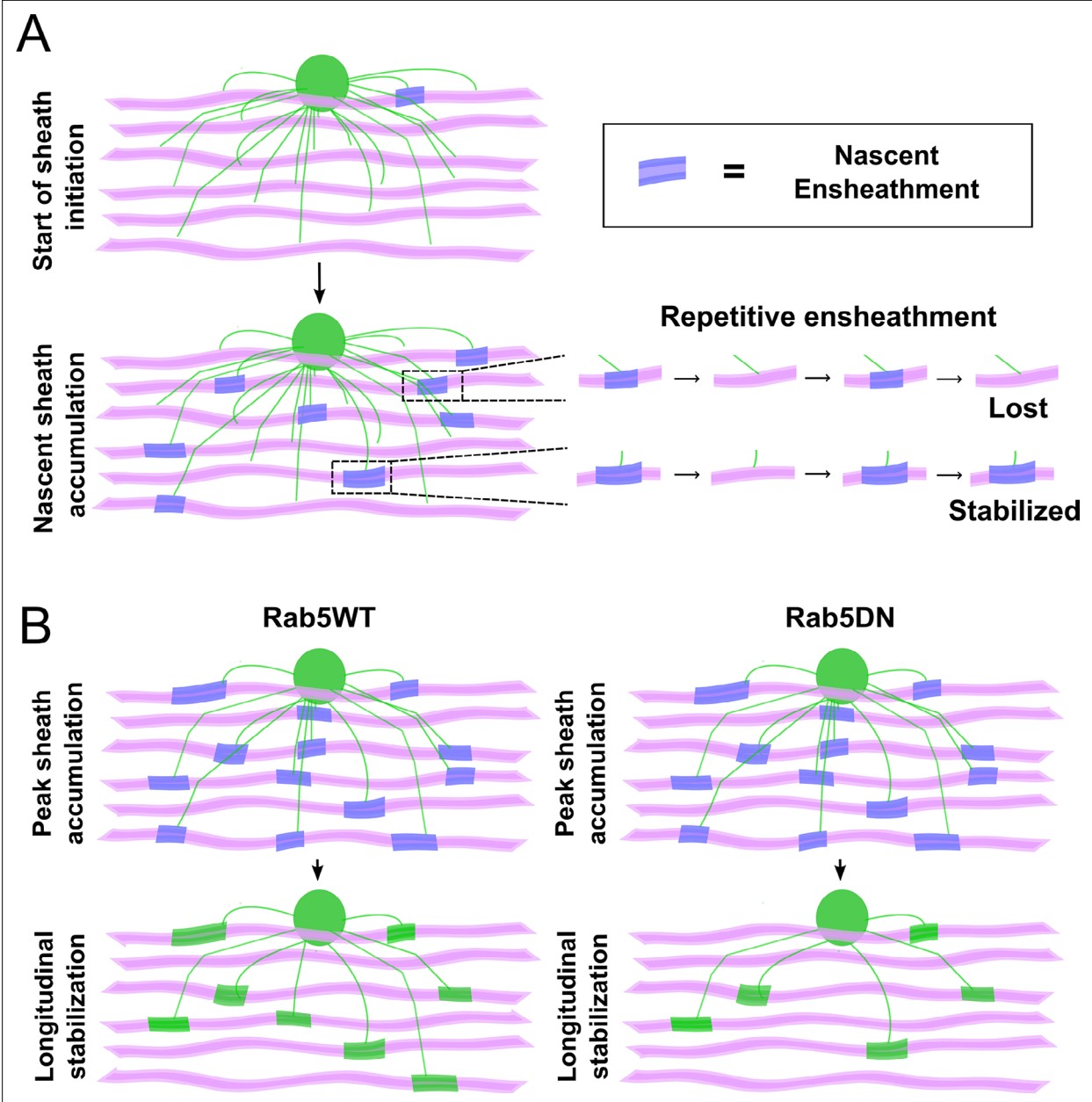

**Figure 10.** The process of sheath accumulation involves the repetitive ensheathment of axons and Rab5 promotes longitudinal sheath stabilization. (**A**) A model depicting how the repetitive ensheathment of axons is occurring during the dynamic accumulation phase of the sheath formation process. Oligodendrocytes are green, axons are pink, and the nascent ensheathments are purple. (**B**) A model depicting that Rab5WT helps to promote the stabilization of nascent sheaths during the stabilization phase of the sheath formation process (left). The Rab5DN mutant instead results in fewer stabilized sheaths (right). Oligodendrocytes are green, axons are pink, nascent ensheathments are purple, and stabilized ensheathments are also green.

accumulate sheaths over the course of a few hours. However, imaging intervals of 10–20 min were used, and sheaths had to be at least 5 µm in length and stable for >10 min to be counted. Our studies show that immature ensheathments as short as 2 µm can wrap around axons, and many of these disappear before growing to 5 µm. To effectively track the dynamics of these initial ensheathments, 5 min was the minimum imaging interval necessary based on our criteria. Our work builds on these earlier studies (*Czopka et al., 2013*; *Mensch et al., 2015*) to show that while oligodendrocytes accumulate sheaths in a relatively short time, this phase is highly dynamic and involves extensive sheath initiation and loss.

Myelin sheath formation can be most simply described as a combination of axon adhesion and membrane biogenesis. So why are so many immature ensheathments lost? Our axonal ensheathment dynamics experiments demonstrated that oligodendrocytes will ensheathe the same domain of an axon an average of 2–3 times before a stable sheath is formed. This extensive 'sampling' of axons plausibly finds regions that are ready to be myelinated. Successful sheath formation presumably occurs on axons that are primed with the appropriate adhesion molecules needed to facilitate binding and downstream wrapping. Previous work supports this idea, showing that the disruption of different adhesion molecules in both neurons and oligodendrocytes can result in reduced myelin sheath number (*Hughes and Appel, 2019*; *Elazar et al., 2019*; *Djannatian et al., 2019*; *Bartsch et al., 1997*). To further build on this concept, axonal vesicular release could be a part of this priming mechanism. This is suggested by the observation that synaptic adhesion proteins accumulate underneath and around myelin sheaths in axons to promote stability and sheath growth (*Hines et al., 2015*; *Mensch et al., 2015*; *Hughes and Appel, 2019*; *Almeida et al., 2021*). Recent evidence even suggests that myelin stimulates this vesicular release as part of a feed-forward mechanism to promote myelination (*Almeida et al., 2021*). It is tempting to speculate that the repetitive ensheathment of the same domain of an axon could be part of a mechanism for priming axons. However, our activity experiment in *Figure 3* did not significantly impact the repetitive ensheathment of axons. Therefore, it seems reasonable to conclude that activity-dependent axonal vesicular release likely plays a larger role in the subsequent phase of sheath stabilization.

Our work supports a model where Rab5+, Rab7+, and Rab11+ endocytic endosomes localize to immature ensheathments and that Rab5 regulates the long-term stabilization of sheaths. It seems reasonable that the endocytic recycling pathway could be regulating the localization of adhesion proteins within myelin to help optimize sheath stability. The decrease in sheath stability that we observe when inhibiting Rab5 is consistent with some of the phenotypes reported from disrupting adhesion proteins in myelin (*Hughes and Appel, 2019*; *Elazar et al., 2019*; *Djannatian et al., 2019*; *Bartsch et al., 1997*). Previous work showing that the endocytic recycling pathway is critical for the proper localization of myelin proteins such as PLP in oligodendrocyte cultures also supports this idea (*Trajkovic et al., 2006*; *Winterstein et al., 2008*; *Baron et al., 2015*; *Krämer et al., 2001*; *Feldmann et al., 2011*). Our study suggests that the recycling of adhesion molecules or other myelin proteins may be most prevalent during the stabilization phase of sheath formation. However, since we over-expressed a dominant-negative mutant to inhibit Rab5 activity using *myrf*, which is a specific but weak driver, it is possible that expression levels of the mutant were insufficient to disrupt the early accumulation phase of sheath formation. We therefore cannot exclude the possibility that the recycling of adhesion proteins is also important during the early sheath initiation phase.

It is very intriguing that there are regional differences in the number of ensheathments formed by each oligodendrocyte. Dorsal oligodendrocytes are less abundant but produce more ensheathments per cell than ventral oligodendrocytes. Despite these differences, cells from both regions exhibit a similar ratio of ensheathments formed to ensheathments lost. It is possible that oligodendrocytes balance total ensheathment attempts with the rate of stabilization based on the size of the axons that are being wrapped. Cells wrapping large caliber axons might not be able to support as many total sheaths due to the increased amount of membrane required to wrap larger axons. Whether there is a difference in the average axon diameter between the dorsal and ventral regions of the spinal cord is unclear. The mechanism by which oligodendrocytes would sense the caliber of each axon is also speculative, but one could envision that oligodendrocytes might repetitively ensheathe the same axons as a measuring mechanism. However, our data did not find a strong correlation between axon diameter and the number of ensheathment attempts (*Figures 2 and 3*, *Figure 3—figure supplement 3*).

This study highlights several striking observations regarding how oligodendrocyte sheath number is regulated at both the cellular and molecular level. There is a clear relationship between the number of ensheathment attempts performed by an oligodendrocyte and the number of these ensheathments that are ultimately stabilized during development. The endocytic recycling pathway appears to promote the stability of sheaths, but future work is needed to clarify its role. Overall, the current studies will lead to important future research examining how the mechanisms regulating membrane biogenesis and sheath stability cooperate to determine sheath number, sheath thickness, and length.

# Materials and methods

All source data have been provided within this manuscript. All reagents will be made available upon reasonable request to Dr. Wendy Macklin.

## Key resources table

| Reagent type (species) or resource | Designation | Source or reference | Identifiers | Additional information |
|---|---|---|---|---|
| Gene (*Danio rerio*) | RAB5C | Addgene | 80518 | Plasmid |
| Gene (*Danio rerio*) | RAB7A | Addgene | 80522 | Plasmid |
| Gene (*Danio rerio*) | RAB11A | Addgene | 80529 | Plasmid |
| Genetic reagent (*Danio rerio*) | sox10:eGFP-CAAX | This paper | | Expresses membrane localized eGFP in oligodendrocyte lineage. |
| Genetic reagent (*Danio rerio*) | sox10:mScarlet-CAAX | Gift from Appel lab | | Expresses membrane localized mScarlet in oligodendrocyte lineage. |
| Genetic reagent (*Danio rerio*) | neuroD:tagRFP-CAAX | This paper | | Expresses membrane localized tagRFP in neurons. |
| Genetic reagent (*Danio rerio*) | myrf:eGFP-RAB5C | This paper | | Expresses eGFP-Rab5 fusion in pre-myelinating oligodendrocytes. |
| Genetic reagent (*Danio rerio*) | myrf:eGFP-RAB7A | This paper | | Expresses eGFP-Rab7 fusion in pre-myelinating oligodendrocytes. |
| Genetic reagent (*Danio rerio*) | myrf:eGFP-RAB11A | This paper | | Expresses eGFP-Rab11 fusion in pre-myelinating oligodendrocytes. |
| Genetic reagent (*Danio rerio*) | myrf:tagRFP-RAB5C | This paper | | Expresses tagRFP-Rab5 fusion in pre-myelinating oligodendrocytes. |
| Genetic reagent (*Danio rerio*) | myrf:tagRFP-RAB7A | This paper | | Expresses tagRFP-Rab7 fusion in pre-myelinating oligodendrocytes. |
| Genetic reagent (*Danio rerio*) | myrf:tagRFP-RAB11A | This paper | | Expresses tagRFP-Rab11 fusion in pre-myelinating oligodendrocytes. |
| Genetic reagent (*Danio rerio*) | myrf:tagRFP-rab5C$^{S36N}$ | This paper and *Clark et al., 2011* | | Expresses tagRFP-Rab5 DN mutant fusion in pre-myelinating oligodendrocytes. |
| Genetic reagent (*Danio rerio*) | myrf:tagRFP-rab7A$^{T22N}$ | This paper and *Clark et al., 2011* | | Expresses tagRFP-Rab7 DN mutant fusion in pre-myelinating oligodendrocytes. |
| Genetic reagent (*Danio rerio*) | myrf:tagRFP-rab11A$^{S25N}$ | This paper and *Clark et al., 2011* | | Expresses tagRFP-Rab11 DN mutant fusion in pre-myelinating oligodendrocytes. |
| Genetic reagent (*Danio rerio*) | myrf:tagRFP | This paper | | Expresses cytosolic tagRFP in pre-myelinating oligodendrocytes. |
| Genetic reagent (*Danio rerio*) | Tg(nkx2.2a:EGFP-CAAX) | *Kirby et al., 2006*; *Kucenas et al., 2008* | | Transgenic line expressing membrane localized eGFP in oligodendrocyte lineage. |
| Genetic reagent (*Danio rerio*) | Tg(mbp:eGFP-CAAX) | *Preston et al., 2019*; *Brown et al., 2021* | | Transgenic line expressing membrane localized eGFP in myelinating oligodendrocytes. |
| Genetic reagent (*Danio rerio*) | Tg(mbp:tagRFP) | *Preston et al., 2019*; *Brown et al., 2021* | | Transgenic line expressing cytosolic tagRFP in myelinating oligodendrocytes. |
| Genetic reagent (*Danio rerio*) | Tg(elav3:H2B-GCaMP6f) | Gift from Dr. David Schoppik *Dunn et al., 2016* | | Transgenic line expressing nuclear localized GCaMP6f in neurons. |
| Chemical compound, drug | Tricaine methanesulfonate | Syndel | MS-222 | Anesthesia |
| Chemical compound, drug | Pancuronium bromide | Sigma | P1918 | Anesthesia |

## Zebrafish lines and husbandry

The Institutional Animal Care and Use Committee at the University of Colorado School of Medicine approved all animal work (#00419). This group follows the U.S. National Research Council's Guide

for the Care and Use of Laboratory Animals and the U.S. Public Health Service's Policy on Humane Care and Use of Laboratory Animals. Zebrafish larvae were raised at 28.5°C in embryo medium and were staged as dpf. Transgenic lines used in this study were *Tg(nkx2.2a:EGFP-CAAX)*, *Tg(mbp:eGFP-CAAX)*, and *Tg(mbp:tagRFP)* (*Kirby et al., 2006*; *Kucenas et al., 2008*; *Preston et al., 2019*; *Brown et al., 2021*). The *Tg(elav3:H2B-GCaMP6f)* line was a kind gift from Dr. David Schoppik (*Dunn et al., 2016*). This line expresses GCaMP6f fused to the histone H2B and is thus localized to the nucleus. All other constructs were introduced by transient transgenesis to achieve sparse labeling for single-cell analysis in wild-type ABs. All experiments and analyses were performed blind to gender since sex is determined at later stages than those studied here.

## Plasmid construction

Multi-site gateway cloning was used to produce the following plasmids: *pEXPR-sox10:eGFP-CAAX*, *pEXPR-sox10:mScarlet-CAAX* (gift from Bruce Appel's lab), *pEXPR-neuroD:tagRFP-CAAX*, and *pEXPR-mbp:eGFP*. The first three plasmids contain a cysteine-aliphatic amino acid-X (CAAX) prenylation motif that targets the fluorescent protein to the cell membrane. The following entry plasmids were used to build these constructs: *p5E-sox10* (*Mathews and Appel, 2016*), *p5E-neuroD* (gift from Bruce Appel's lab), *p5E-mbp* (*Brown et al., 2021*), *pME-eGFP-CAAX*, *pME-tagRFP-CAAX*, *p3E-eGFP*, *p3E-polyA*, *pDEST-Tol2* (no transgenesis marker).

The *pEXPR-mbp:eGFP* plasmid was linearized using *Sal*I and *SnaB*I to remove both the *mbp* driver and *eGFP* insert. An *myrf* driver element was PCR amplified from *p5E-myrf* (gift from Bruce Appel's lab) and was ligated into the linearized vector with an *eGFP* insert flanked with *BamH*I and *Age*I sites using NEBuilder HiFi assembly cloning. The resulting *pEXPR-myrf:eGFP* plasmid was confirmed by diagnostic digest and sequencing and was used as a base vector for building the remaining over-expression constructs.

Plasmids encoding the *RAB5C*, *RAB7A*, and *RAB11A* zebrafish coding sequences (Addgene, *RAB5C*=80518, *RAB7A*=80522, and *RAB11A*=80529) were sequenced, and the *RAB7A* and *RAB11A* sequences each had a point mutation (*D63G* and *Q166R*, respectively) relative to the NCBI protein sequences (*RAB7A*=NM_200928.1, *RAB11A*=NM_001007359.1) and relative to other previous work in zebrafish (*Clark et al., 2011*). These mutations were corrected using a combination of mutagenesis by PCR-driven overlap extension (*Heckman and Pease, 2007*) and NEBuilder HiFi assembly cloning.

The *RAB5C*, *RAB7A*, and *RAB11A* sequences were then cloned directly into the *pEXPR-myrf:eGFP* plasmid as follows. This plasmid was linearized using *BamH*I and *Age*I to remove the *eGFP* insert. Each *RAB* sequence was PCR amplified using the following primer sequences (additional overhang sequences were added to each primer for NEBuilder HiFi assembly-based cloning that are not shown here):

- atggcggggcgaggtgg (Rab5c F)
- ttagtttccgcctccacagc (Rab5c R)
- atgacatcaaggaagaaagttcttctgaagg (Rab7a F)
- tcagcagctacaggtctctgc (Rab7a R)
- atggggacacgagacgacg (Rab11a F)
- ctagatgctctggcagcactgc (Rab11a R)

The *RAB* PCR fragments were then ligated into the linearized plasmid, along with either *tagRFP* or *eGFP* PCR fragments, using NEBuilder HiFi to generate the following fusion constructs (a kozak sequence, GCCACC, was added directly 5' of the translational start site for all constructs):

- *pEXPR-myrf:eGFP-RAB5C*
- *pEXPR-myrf:eGFP-RAB7A*
- *pEXPR-myrf:eGFP-RAB11A*
- *pEXPR-myrf:tagRFP-RAB5C*
- *pEXPR-myrf:tagRFP-RAB7A*
- *pEXPR-myrf:tagRFP-RAB11A*
- *pEXPR-myrf:tagRFP*

We also made a set of dominant-negative point mutations for each of the *RAB* genes ($rab5C^{S36N}$, $rab7A^{T22N}$, $rab11A^{S25N}$) that have been characterized in zebrafish previously (*Clark et al., 2011*). We

generated the *rab5C^{S36N}* mutant through a combination of mutagenesis by PCR-driven overlap extension (*Heckman and Pease, 2007*) and NEBuilder HiFi assembly cloning. Primer sequences:

- *rab5C^{S36N}*
- PCR fragment 1, mutation site at 3' end of R2 primer.
  - atggcggggcgaggtgg (F)
  - cagactctcccagcaacacaagc (R1)
  - ccaggctgttcttgcctactgcagactctcccagcaacacaagc (R2)
- PCR fragment 2, mutation site at 5' end of F2 primer.
  - tgctgcgcttcgtcaaaggc F1
  - cagtaggcaagaacagcctggtgctgcgcttcgtcaaaggc (F2)
  - ttagtttccgcctccacagc (R)

Both final PCR fragments were then ligated into the same linearized *pEXPR:myrf* plasmid described above, along with a *tagRFP* PCR fragment, using NEBuilder HiFi to generate the following fusion plasmid (a kozak sequence, GCCACC, is directly 5' of the translational start site):

- *pEXPR-myrf:tagRFP-rab5C^{S36N}*

The *rab7A^{T22N}* and *rab11A^{S25N}* mutations were made by Keyclone Technologies (San Marcos, CA, USA). Resulting plasmids:

- *pEXPR-myrf:tagRFP-rab7A^{T22N}*
- *pEXPR-myrf:tagRFP-rab11A^{S25N}*

All plasmids were confirmed by diagnostic restriction digest and sequencing.

## Injections, anesthesia, mounting, and general imaging parameters

Plasmids were injected into zebrafish embryos at the single-cell stage with Tol2 mRNA to achieve transient transgenesis and sparse labeling. On the desired day, the injected larvae were mounted laterally in 0.8% low-melt agarose with 140 µg/mL (0.014%) of tricaine methanesulfonate (Syndel) as anesthesia. All imaging was performed live using a Nikon A1R resonance scanning confocal microscope and a 40× Apochromat long-working distance water immersion objective with a 1.15 NA (Nikon MRD77410). A temperature-controlled stage maintained at 28.5°C was used for all time-lapse experiments. All imaging was done in the spinal cord of each larva above the yolk-sac extension. Oligodendrocytes in the ventral spinal cord that myelinated Mauthner axons were excluded from all analyses in this work.

In the axonal ensheathment activity experiment in *Figure 3*, we compared tricaine and pancuronium bromide (Sigma, P1918) forms of anesthesia. At 2.5 dpf, larvae were put down with tricaine and a tiny slit was made at the tip of each tail using a razor blade. Some of these larvae were then mounted in agarose containing the same concentration of tricaine used in all other experiments (no activity group) and the rest of the larvae were transferred into embryo media containing 0.45 mg/mL pancuronium bromide and without tricaine for 5 min. These larvae were then mounted in agarose containing the same concentration of pancuronium bromide (activity group). All other general imaging parameters were kept the same for this experiment.

For the calcium activity experiment, *Tg(elav3:H2B-GCaMP6f)* larvae were all put down in pancuronium bromide exactly as above. (See this experimental section below for additional details of how the tricaine spike was performed.)

## Myelinating oligodendrocyte cell counts in the spinal cord

*Tg(mbp:eGFP-CAAX)* and *Tg(mbp:tagRFP)* transgenic lines that express membrane-tethered eGFP and cytosolic tagRFP in myelinating oligodendrocytes respectively were crossed and embryos were grown to 4 dpf. Both sides of the spinal cord for each larva were imaged laterally above the yolk-sac extension with a 1× optical zoom (0.31 µm XY pixel size), a 0.3 µm z-step size (Nyquist), and 32× line averaging. Cell counts were performed in Imaris (version 9.8). Each image was cropped to separate dorsal and ventral regions. A Gaussian filter was then applied with a 0.311 µm width to smooth out noise. Following this, a local-background subtraction was performed with an estimated cell body size of 7 µm. A threshold for fluorescent intensity was then applied to each image manually. Finally, a 0.5 µm water-shed filter was used to separate contacting cell bodies and Imaris counted the number

of cells in each image. Cell bodies occupying the area in between the dorsal and ventral axon tracts were manually removed (these cells were in the minority).

## Static oligodendrocyte imaging for sheath number/length analysis

Individual sheaths were visualized by labeling oligodendrocytes with *sox10:eGFP-CAAX* by transient transgenesis. These cells were imaged at 4 dpf with a 2× optical zoom (0.16 μm XY pixel size), a 0.3 μm z-step size (Nyquist), and 32× line averaging. Cells were chosen that were not too crowded for accurate quantification. Except for *Figure 1D*, dorsal and ventral cells were never combined for this analysis. In each image every sheath was identified and counted by inspecting individual optical sections. The lengths of each sheath were measured from maximum intensity projections using the simple neurite tracer plugin in Fiji. From this data we calculated sheath number, average sheath length, and the total sheath length per cell (calculated by adding up the length of all sheaths supported by each cell). The same images and ventral control data from this analysis were used and presented in *Figures 1 and 8*, and *Figure 4—figure supplement 1*.

## Axonal ensheathment dynamics imaging and analysis

Criteria for quantifying ensheathment attempts using labeled axons were established. Axon membranes were sparsely labeled with the *neuroD:tagRFP-CAAX* plasmid by transient transgenesis in the *Tg(nkx2.2:eGFP-CAAX)* stable line. At 2.5 dpf larvae were anesthetized in tricaine and mounted, and time-lapse imaging was performed for 15–18 hr with an imaging interval of 5 min. A 2× optical zoom (0.16 μm XY pixel size), a 0.5 μm z-step size, and 16× line averaging were used. Imaging was done in the spinal cord above the yolk sac extension and focused on dorsal and midline axons, since the *Tg(nkx2.2:eGFP-CAAX)* line was too crowded in the ventral region for visualizing ensheathment dynamics. Regions in each video with potential axonal ensheathments were cropped and corrected for drift using the 3D drift correct plugin in Fiji. Individual optical sections were inspected and a volume projection in Imaris was analyzed for each axonal ensheathment. From this, we developed quantification criteria. Ensheathments had to be ~2 μm or longer and cylindrical in shape around the axon, that is, the axon fluorescent signal had to go through the center of the oligodendrocyte signal, and it had to be more than ~75% of the way around the axon. Once an ensheathment had formed, it had to shrink to below 2 μm in length and be less than half-way around the axon to be considered lost.

Repetitive ensheathments were multiple rounds of sheath initiation/loss on the same domain of an axon. Each axon domain was the region between the two most lateral ensheathment attempts that took place during a series of repetitive ensheathments. In cases with only a single ensheathment attempt, the axon domain was the region directly underneath the sheath. These domains ranged from ~2 to 20 μm in length. Some axons had more than one domain that was ensheathed. These domains were considered separate because they were ensheathed by different oligodendrocytes or because a single oligodendrocyte had more than one sheath on the same axon at the same time. Axonal domains were considered destabilized if each ensheathment was lost by the end of the time-lapse period and stable if they remained. We required that a minimum of 3 hr of video be acquired after the final ensheathment attempt to be quantified.

For the activity axonal ensheathment experiment in *Figure 3*, *Tg(nkx2.2:eGFP-CAAX)* larvae were anesthetized and mounted in either tricaine or pancuronium bromide as described in the 'Injections, anesthesia, mounting, and general imaging parameters' section above. Using multi-well glass-bottom dishes (Greiner Bio-One, 627871), we imaged larvae from each group in each experiment exactly as described earlier in this section, and the videos were all processed and quantified exactly as was described.

We present data in *Figure 3—figure supplement 2C, D* showing how often the same process from the same oligodendrocyte performs each repetitive ensheathment. We also report on how often these processes seem to remain in contact with the axon. Except for one instance in this data set (64 repetitively ensheathed domains), if the same process performed each repetitive ensheathment it also remained in contact with the axon. For instances where different processes performed the repetitive ensheathments, this involved complete process retraction before a different process started the next ensheathment in the series. About 15–20% of processes could not be sufficiently tracked throughout the entire video due to periods of crowding from other processes. The activity for these instances was categorized as 'unclear'. It is important to note that our imaging interval for these experiments was

5 min, which is not fast enough to fully capture all oligodendrocyte process dynamics. Analysis of this data set therefore could have missed some of these process movements.

## Calcium imaging and quantification

*Tg(elav3:H2B-GCaMP6f)* larvae were mounted at 2.5 dpf in pancuronium bromide as described in the 'Injections, anesthesia, mounting, and general imaging parameters' section above. We mounted two to three larvae in each well of a multi-well glass-bottom dish (Greiner Bio-One, 627871). This allowed us to pick a single larva per well to do a spike experiment on and then multiple spike experiments could be performed per dish. The same region of the spinal cord above the posterior portion of the yolk-sac extension was imaged. We used a 1× optical zoom (0.31 µm XY pixel size) and 8× line averaging. A 5 min, single optical-section, time-lapse video was collected at 4 Hz (4 fps) to get a baseline of calcium transients in each larva. Each well had 500 µL of embryo media with 0.45 mg/mL pancuronium bromide (same concentration as the agar) for the initial round of imaging. After this, each well was spiked with 500 µL of embryo media containing either pancuronium bromide (control group) or tricaine. For the control group the embryo media contained the same concentration of pancuronium bromide (0.45 mg/mL) and the embryo media for the tricaine spike group was at a 2× concentration (140 µg/mL final concentration as was used for all other experiments). All larvae were incubated for 5 min after the spike. Then, a final round of imaging was performed on the same cells in the same field of view for measuring changes in the overall number of transients.

All videos were processed and quantified post blinding. ROIs for the neuronal cell bodies in each field of view were generated in Fiji. The 'despeckle' filter was first used to clean up salt-and-pepper noise. A standard deviation time projection was made to create a visual of all the cell bodies in each video. Huang's threshold was then applied manually to each projection to binarize the signal of each cell body. We used the 'analyze particles' function to automate the generation of ROIs around each cell. The 'water-shed' function was then used to separate cell bodies that were touching. Each set of ROIs were then applied to the raw video file and additional ROIs were added manually as needed that were missed by the segmentation work-flow. The mean intensity within each ROI was collected for every frame of the video and was transferred to excel for further analysis. Two ROIs were placed manually in each video for determining background levels. All the intensities in every frame for both background ROIs were averaged, and this single value was subtracted from every other mean intensity value that was collected. Calcium transients were determined with a moving baseline. Ten frames of video prior to each transient were averaged to set the baseline. The fluorescence intensity had to be 3 standard deviations above this baseline for a minimum of three frames to be counted. Within complex calcium transients with multiple peaks, transients were counted as being separate if the signal returned to below 3 standard deviations before going back up again. The total number of calcium transients within each video was quantified and after unblinding, the pre- and post-calcium transient numbers were compared.

## Axon diameter measurements

The diameters of all the domains in both axonal ensheathment experiments (*Figures 2 and 3*) were measured by selecting the first video frame of a single ensheathment for each axonal domain; a single optical section was then isolated for measurement in Fiji. Two different line scans with a 3-pixel width were drawn manually across each axon at the ensheathment site. The axon fluorescence intensity data was transferred to excel for further analysis. Two 8×4-pixel ROIs were also drawn to measure the background intensity. After background subtraction, the full-width 1/3 maximum was determined for each line scan manually and the two values from each domain were averaged as the representative diameter of each axonal domain.

## Oligodendrocyte ensheathment dynamics imaging and analysis

Oligodendrocytes were sparsely double-labeled with our *pEXPR-sox10:eGFP-CAAX* plasmid and one of the following plasmids using transient transgenesis: *pEXPR-myrf:tagRFP, pEXPR-myrf:tagRFP-RAB5C*, or *pEXPR-myrf:tagRFP- rab5C^{S36N}*. The *myrf*-driven plasmids identified cells that were fated to make myelin. At 2.5 dpf, larvae were anesthetized in tricaine and mounted, and eGFP+/RFP+ progenitor cells were chosen for imaging above the yolk-sac extension in the spinal cord. Time-lapse imaging was performed for 15 hr with an imaging interval of 5 min. A 2× optical zoom (0.16 µm XY pixel size),

a 0.75 µm z-step size, and 16× line averaging were used. After the time-lapse imaging period was over at 3 dpf, the larvae were removed from the mounting agar and were put back into embryo medium in separate wells of a 24-well plate. At 4 dpf, a final static image was taken for each of the same cells to conclude the imaging paradigm. To be considered for analysis, we had to obtain at least 30 min of imaging before each cell started the ensheathment process. Additionally, each cell had to stop making sheaths for at least 90 min before the end of the imaging interval.

A max projection of each video was background-subtracted and corrected for drift using Fiji's 3D drift correct plugin. Sheath initiation and loss were counted manually in every frame using the multi-point counter in ImageJ based on the ensheathment criteria we established in the previous section. Although the counter ROIs for this were placed within each max projection, every sheath initiation/loss event was determined by inspecting individual optical sections and by looking at a volume projection in Imaris for each frame. At the final 4 dpf time point, all sheaths were counted, and the lengths were measured using the simple neurite tracer plugin in Fiji.

A quality control experiment determined that the conditions of this imaging paradigm did not significantly change the average sheath number or sheath length across a population of cells. Static images of both dorsal and ventral oligodendrocytes were collected at 4 dpf from zebrafish larvae that had been anesthetized in tricaine, mounted in agar, and housed in the time-lapse imaging chamber for the 15 hr experiment from 2.5 to 3 dpf. However, these larvae were not imaged during this time. We compared the sheath number and lengths for these cells (agar_tricaine condition) to our sheath number counts from *Figure 1C* (standard condition) and to cells that went through the full ensheathment dynamics imaging paradigm (time-lapse condition) (*Figure 4—figure supplement 1A–D*).

A separate quality control experiment determined that oligodendrocytes did not make any new ensheathments during the stabilization phase from 3 to 4 dpf. Cells were imaged using a modified ensheathment dynamics imaging paradigm. First, from 2.5 to 3 dpf, the imaging interval was 30 min rather than 5 min. At 3 dpf, the larvae were removed from the tricaine and imaging agar and were allowed to recover in embryo media for 30 min. After this the larvae were remounted, and the same cells were imaged for an additional 23 hr with an imaging interval of 15 min. We tracked sheath initiation and loss from 3 to 4 dpf for five dorsal cells and three ventral cells. None of these eight oligodendrocytes produced any new sheaths from 3 to 4 dpf (*Figure 6—figure supplement 1*).

## Endosome imaging and quantification

Our *myrf:eGFP-RAB5C, -RAB11A,* and *-RAB7A* fusion constructs were transiently expressed alongside *sox10*:mScarlet-CAAX to label the membrane of individual oligodendrocytes. At 2.5 dpf, larvae were anesthetized in tricaine, mounted, and cells that were in the early stages of the ensheathment process were chosen for imaging. Static snapshots of a mix of both dorsal and ventral cells were taken using a 3× optical zoom (0.1 µm XY pixel size, Nyquist), a 0.75 µm z-step size, and 16–32× line averaging. A larger z-step size than Nyquist (0.3 µm) was used to minimize motion artifacts during imaging. Images were pre-screened by viewing the membrane marker channel only and individual sheaths with high-quality imaging were cropped for further processing. The number of endosomal puncta in each image was quantified using a spots analysis in Imaris (version 9.2). We used an estimated XY diameter of 0.5 µm to detect Rab5 and 0.4 µm to detect Rab7 and Rab11. After a local-background subtraction, a threshold for fluorescent intensity was applied to each image manually. Imaris counted the number of puncta, and we normalized these numbers to the length of each sheath (in µm), measured using the Imaris filament tracer. Only puncta within a sheath were counted based on overlap with the membrane marker. Puncta that were present in adjacent processes or other structures were manually excluded from each image.

## Statistics, sample size determination, and reproducibility/rigor

All statistics and plots were done using GraphPad Prism (version 9). We used the non-parametric Mann-Whitney test for all unpaired comparisons with no assumption of normality. We used the non-parametric Wilcoxon test for all paired comparisons with no assumption of normality. For groups of three or more, global significance was first assessed using the non-parametric Kruskal-Wallis test. This was followed up by Dunn's multiple comparison tests if the global p-value was <0.05 (unless otherwise stated in each figure legend). Global p-values are reported when not significant. Non-parametric statistical tests were chosen since each data set had groups that were not normally distributed and/

or had unequal variances. A Fisher's exact test was used when testing categorical data because our sample sizes were less than 100. Simple linear regressions were performed for comparing the association of two variables with each other. Individual data points are shown for all plots with the central dashed lines representing the average of the group. We considered p < 0.05 the threshold for statistical significance and we provide exact p-values for all analyses.

The sample sizes for the cell counts in *Figure 1*, the repetitive axonal ensheathment analyses in *Figures 2 and 3*, the Rab+ endosomal quantification in *Figure 7*, and the modified ensheathment dynamics data in *Figure 6—figure supplement 1* were not pre-determined. The sample sizes for the sheath analyses performed in *Figures 1 and 8* and *Figure 4—figure supplement 1* were determined based on previous practices and are meant to show the full distribution for this type of data set. The control, Rab5WT, and Rab5DN 'sheath number per cell' data in *Figure 8* was used to do a power analysis (GPower Version 3.1) to determine the appropriate sample sizes for the oligodendrocyte ensheathment dynamics data in *Figures 4–6* and *Figure 9*, and *Figure 4—figure supplement 1* (time-lapse group) and *Figure 9—figure supplement 1*. We performed a power analysis using the parametric 'ANOVA: Fixed effects, omnibus, one-way' stats test in GPower with three groups (control, Rab5WT, Rab5DN) with an alpha error probability of 0.05 and 80% power. The effect size was calculated within GPower. From this, GPower calculated needing 15 cells per group. Since sheath number was not normally distributed, we had to assess statistical significance for these experiments using a non-parametric test. We therefore increased the sample size by 20% (n=15 increased to n=18) to account for the decreased power of non-parametric tests. The same time-lapse ventral control data set was used in *Figures 5, 6 and 9*, and *Figure 4—figure supplement 1* because this data is very difficult to collect, and the analysis was very time intensive. Therefore, we also used the results of this power analysis to determine the same sample size of the dorsal group of cells analyzed in *Figures 4–6* and *Figure 4—figure supplement 1* (time-lapse group).

All representative images in each figure were from the associated quantified data set. These images were adjusted for brightness and contrast for clarity of presentation. No quantifiable data was excluded from any analysis. Results/phenotypes that were replicated are present within this manuscript. All data in each figure was pooled from a minimum of two independent rounds of fish crossing and injections. Data for each individual oligodendrocyte in this study is considered a biological replicate because only one cell was imaged from each larva unless otherwise noted in each figure legend. Each individual axon in the axonal ensheathment dynamics experiments was considered a biological replicate. Individual larvae were biological replicates for cell counting and for the calcium imaging experiment. Data points collected from the same cell (different sheath lengths from the same oligodendrocyte, multiple domains on the same axon, rab puncta counts in individual sheaths) are considered as technical replicates in this study. All analyses were performed blind using a custom Fiji script called 'blind-files' (source code can be found at https://github.com/Macklin-Lab/imagej-microscopy-scripts, copy archived at *George, 2019*).

## Acknowledgements

This work was supported by National Institutes of Health R37NS82203 to WBM and NIH 5F31NS118830 to ARA. We thank Dr. Andrew Lapato and Dr. Caleb Doll for valuable discussion and feedback on this manuscript.

## Additional information

### Funding

| Funder | Grant reference number | Author |
| --- | --- | --- |
| National Institutes of Health | 5F31NS118830 | Adam R Almeida |
| National Institutes of Health | R37NS82203 | Wendy B Macklin |

| Funder | Grant reference number | Author |
|---|---|---|

The funders had no role in study design, data collection and interpretation, or the decision to submit the work for publication.

## Author contributions

Adam R Almeida, Conceptualization, Data curation, Formal analysis, Funding acquisition, Validation, Investigation, Visualization, Methodology, Writing – original draft, Writing – review and editing; Wendy B Macklin, Conceptualization, Resources, Data curation, Supervision, Funding acquisition, Validation, Visualization, Methodology, Project administration, Writing – review and editing

## Author ORCIDs

Adam R Almeida http://orcid.org/0000-0001-9249-6688
Wendy B Macklin http://orcid.org/0000-0002-1252-0607

## Ethics

The Institutional Animal Care and Use Committee at the University of Colorado School of Medicine approved all animal work (#00419). This group follows the U.S. National Research Council's Guide for the Care and Use of Laboratory Animals and the U.S. Public Health Service's Policy on Humane Care and Use of Laboratory Animals.

## Decision letter and Author response

Decision letter https://doi.org/10.7554/eLife.82111.sa1
Author response https://doi.org/10.7554/eLife.82111.sa2

## Additional files

### Supplementary files

- MDAR checklist

### Data availability

All datasets generated and analyzed during this study are included in the manuscript.

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
