## [Editor Report]

The study by Almeida and Macklin takes advantage of the power of zebrafish as a model system to study the dynamic behavior of myelinating oligodendrocytes in a living system. Using a combination of time-lapse imaging and transgenic constructs in larval zebrafish, the authors compellingly demonstrate that oligodendrocytes balance sampling of axons with stabilization of myelin sheaths at a stereotypical rate during development. The study provides valuable new insights into how oligodendrocytes undertake myelination and will be useful to the field.

---

## [Decision Letter]

**Decision letter after peer review:**

Thank you for submitting your article "Early myelination involves the dynamic and repetitive ensheathment of axons which resolves through a low and consistent stabilization rate" for consideration by *eLife*. Your article has been reviewed by 3 peer reviewers, and the evaluation has been overseen by a Reviewing Editor and Marianne Bronner as the Senior Editor. The following individuals involved in review of your submission have agreed to reveal their identity: Robert H Miller (Reviewer #1); Sarah C Petersen (Reviewer #3).

Essential revisions:

Experimental

1. Clarification of cellular behaviors will be required. Are the oligodendrocytes wrapping and unwrapping while maintaining some axonal contact or are the oligodendrocytes completely withdrawing from the axon? The authors should analyze their existing data to clarify this point.

2. Dynamic events from movies in which only oligodendrocytes are labeled should be re-analyzed to further characterize the dynamic events. Specifically, how many events involve processes in contact with axons or very close to axons vs. how many events involve process extension and retraction?

3. Does transient disruption of actin polymerization or microtubule assembly affect early OPC/neuron interactions?

4. Axon diameter should be quantified and correlated with data in hand to determine if diameter biases sampling and stabilization of myelin sheaths.

5. Apart from Figure 2, it was not clear from the data provided that the same axon is being contacted. This should be shown more convincingly or the text should be updated accordingly.

6. Assessing the impact of neuronal activity in this system would strengthen the work.

7. The Rab5 data and discussion in the manuscript were disjointed as presented. At minimum, clearer data descriptions and enhanced discussion would better unify the work, but the manuscript would be strengthened with further experimentation using an earlier promoter or by extending the Rab5-DN studies.

Text/figure changes

1. Representative movie/video files should be included to accompany still images shown in figures.

2. Text should be modified to clarify that oligodendrocytes can make a single myelin sheath and that such cells were not studied in detail in the present manuscript.

3. Rab fusion protein images are difficult to interpret and it is unclear how the overexpression constructs relate to endogenous localization. If endogenous labeling could be performed, this would be ideal, but at the very least the authors should provide higher resolution images of the data in hand.

4. The authors should double check the statistical methods employed in Figure 7 and clarify statistics throughout the manuscript to better describe the results.

*Reviewer #1 (Recommendations for the authors):*

There are a number of issues that should be addressed to strengthen the manuscript as currently written.

1. Based on the data provided the studies are heavily overinterpreted and the language suggests a more profound result than is actually presented. It is not clear that the retraction of processes represents a destabilization of a complex sheath structure, and the text should be adjusted accordingly.

2. Along the same lines, OPCs are highly motile and time-lapse studies of oligodendrocytes in vitro show extensive process extension and retraction: it seems likely the authors are seeing the same thing in vivo. It might be interesting to see if transient disruption of actin polymerization or microtubule assembly affected the nature of these early interactions.

3. The repeated extension/retractions may reflect maturation of the oligodendrocytes and the random association of the nearest target axon to the process tip rather than the proposed molecular tuning. It was not clear the same axon was always targeted. While the data in Figure 2 shows the same axon being contacted, it is less clear in the other figures that it is always the exact same axon. The authors should clarify this point.

4. It is not clear how perturbation of Rab5 results in a loss of sheath stabilization, and Figure 6A seems redundant. This component of the paper seems somewhat out of line with the rest of the manuscript and maybe some additional studies – or even enhanced discussion would help with unification.

5. The studies would be strengthened by some perturbation studies. Given the short time frame of the initial cell interaction studies would TTX treatment be feasible to see if there is a role for activity in the selection process?

6. Does the difference in sheath number between dorsal and ventral cells reflect axon diameter with more smaller sheaths generated in dorsal regions, or is it a reflection of different populations of oligodendrocytes? Some additional quantification or at least discussion would be helpful.

*Reviewer #2 (Recommendations for the authors):*

There is a wealth of data underpinning this manuscript, particularly with respect to the time-lapse datasets of myelination by oligodendrocytes in the zebrafish spinal cord. Although much of the data align well with previous analyses in the system, some novel insights are documented. The principal new finding is that of the repetitive ensheathment and unwrapping of early myelinating processes from single axons, and the apparently very high rate at which this occurs. I am, however, a little unclear as to how the various analyses in the manuscript relate to one another and think that a few aspects need to be clarified to allow the clearest conclusions to be drawn.

1. In Figure 2, there are very nice reconstructions of early ensheathing processes contacting and at least partly enwrapping axons, and exhibiting this wrapping and unwrapping type behaviour. However, what is not clear to me from the analysis presented is to what extent the unwrapping and re-wrapping represents a complete withdrawal from the axon or simply an unwrapping while retaining contact with the axon. It appears from the images shown that this wrapping-unwrapping-wrapping appears to show the myelinating process remains in continuous contact with the axon. One would suspect that the cell biology underlying a local wrapping-unwrapping-wrapping phenomenon, while maintaining contact with the axon to be quite distinct from making contact, wrapping, unwrapping, completely withdrawing from the axon, re-extending a process, re-wrapping etc. The data are clearly to hand, and I think an even more detailed description of the phenomena could be presented. It could obviously be that some processes do the former and others the latter, in which case it would be good to know the relative proportion doing so. This seems like an important additional analysis to me.

2. A huge amount of data centres on the analysis of time-lapse datasets where only the oligodendrocytes are labelled. I realise that this means that the authors can't be as definitive about the wrapping-unwrapping-wrapping of individual axons, but they could presumably tell which ensheathment events happen in the same region very locally, and which involve clear extension of processes and clear retraction of ensheathments from the axon entirely. It is basically the same point as above, but I think it would be very useful to categorise these very dynamic events further so that the field could better understand if we were observing dynamics at the level of processes in contact with axons/ very close to axons, or dynamic events that involved process extension and retraction as well. I hope this my point on this distinction is both clear to the authors and that they see that it will be important to deconstruct to fully appreciate the dynamics of myelination being described.

3. The data regarding the rab manipulations are interesting, but are clearly somewhat disconnected from the very early dynamics of initial ensheathment that represent arguably the most novel insights. I think that the authors could be a little clearer (relates to points 1 and 2) to make it very clear when during the entire process of myelination the rab effects are pronounced. This is reasonably clear when one digs into the data, but the abstract could be read in such a way that one might think that the various uses of the word stabilisation were of one event, where the rab manipulations made here are likely affecting a later stage than the initial part of the paper. Indeed, it could be that the expression of the constructs using a myrf sequence might have made it very hard to find a phenotype at the earlier initial ensheathment phase, so there is a chance of this being a false negative result. Could an alternative driver (olig1?) be used to test one of the candidates to better match the first part of the manuscript. The authors do discuss the limitations of the manipulation, but perhaps this could be a revision experiment worth the effort to have a more streamlined dataset.

*Reviewer #3 (Recommendations for the authors):*

I suggest the following major experiments:

1) Quantify axonal diameter (from Figure 2) to test whether this biases sampling and ultimately stabilization of myelin sheaths.

2) Similarly, test for effect of neuronal activity and/or subtype in early sampling and stabilization. This could be done in a number of ways. For instance, in Figure 2, distinguishing reticulospinal (phox2b+, activity dependent) or CoPA (tbx16+, activity independent) axons could permit measuring propensity for ensheathment attempt and/or stabilization. Alternatively, neuronal activity could be broadly disrupted or enhanced pharmacologically to test for this effect.

3) Given that already such a small proportion of nascent sheaths are stabilized in wild-type, what happens in larvae with a high number of (either mosaic or stably expressed) Rab5-DN oligodendrocytes? Is there ultimately reduced myelination overall, or is the relatively small decrease (per oligodendrocyte) negligible? This would contextualize the importance of endocytic recycling in myelination.

4) To get at the "why" repeated sampling occurs, the accumulation of synaptic vesicles with (or without) repeated sampling seems an obvious candidate to test. The fact that most destabilized sheaths only sampled the axonal domain once suggests that is insufficient to recruit pro-stabilization factors to the domain, but that 2-4 more sampling attempts could promote SV accumulation.

---

## [Author Response]

Essential revisions:Experimental1. Clarification of cellular behaviors will be required. Are the oligodendrocytes wrapping and unwrapping while maintaining some axonal contact or are the oligodendrocytes completely withdrawing from the axon? The authors should analyze their existing data to clarify this point.

We have performed this analysis using the data in Figures 2 and 3 and the results are presented in Figure 3—figure supplement 2. We have also discussed the results of this analysis on lines 161-181. In the methods, we discuss that our 5-minute imaging interval was designed to visualize the appearance and disappearance of ensheathments. However, this is not fast enough to fully capture the dynamics of each individual process. The analysis we have provided is our best effort from the data that we have (lines 538-547).

2. Dynamic events from movies in which only oligodendrocytes are labeled should be re-analyzed to further characterize the dynamic events. Specifically, how many events involve processes in contact with axons or very close to axons vs. how many events involve process extension and retraction?

This question is best answered from our data where we have axons labeled because it allows us to be more precise regarding the spatial subtleties of what the reviewer is asking. Additionally, the dynamics of each process in our axon experiment is a representation of what each individual process is doing for an entire oligodendrocyte. We have analyzed all the axonal ensheathment data from Figures 2 and 3 which equals a total of n = 91 axonal domains. We believe this data set is large enough to address this question.

3. Does transient disruption of actin polymerization or microtubule assembly affect early OPC/neuron interactions?

We appreciate this important question from the reviewer and are planning to perform these experiments in the future, but it was beyond the scope of the current manuscript.

4. Axon diameter should be quantified and correlated with data in hand to determine if diameter biases sampling and stabilization of myelin sheaths.

This analysis was performed on all the axonal domains from Figures 2 and 3 and is presented in Figure 3—figure supplement 3. We have updated the text in the Results (lines 183-191) and the Discussion (lines 378-386) to include the outcome and interpretation of this data.

5. Apart from Figure 2, it was not clear from the data provided that the same axon is being contacted. This should be shown more convincingly or the text should be updated accordingly.

We appreciate this point from the reviewer. Given the massive number of axons in the tissue, we cannot have individual axons labeled in our individual oligodendrocyte ensheathment dynamics experiments, where we inject to tag single cells. Thus, we do not want to make any claims regarding the repetitive ensheathment of axons in this data. We have updated the text to better clarify this point (lines 212-215).

6. Assessing the impact of neuronal activity in this system would strengthen the work.

This was an excellent suggestion from the reviewers. We performed our time-lapse imaging using tricaine as anesthesia and therefore activity was silenced during these imaging paradigms. We did a quality control experiment in Figure 4.1 supp that showed that keeping larvae in tricaine for 15 hours from 2.5-3dpf did not impact final sheath number at 4dpf. This was our rationale for using tricaine in our experiments. However, it is possible that the repetitive ensheathment phenomenon that we observed in figure 2 was a result of silencing activity. To address this question, we compared tricaine and pancuronium bromide, which does not block CNS activity, in our axonal ensheathment paradigm in Figure 3. The results of this experiment are discussed on lines 132-160 and on lines 358-362 in the discussion.

7. The Rab5 data and discussion in the manuscript were disjointed as presented. At minimum, clearer data descriptions and enhanced discussion would better unify the work, but the manuscript would be strengthened with further experimentation using an earlier promoter or by extending the Rab5-DN studies.

We appreciate this point from the reviewers. We have improved the clarity for the rationale and interpretation of these experiments in our manuscript (Lines 262-267, 300-302, 330-337, and 363-374).

Text/figure changes1. Representative movie/video files should be included to accompany still images shown in figures.

We have submitted these videos with our revisions.

2. Text should be modified to clarify that oligodendrocytes can make a single myelin sheath and that such cells were not studied in detail in the present manuscript.

We have stated this on lines 110-111 and 475-476 to make this very important point more obvious.

3. Rab fusion protein images are difficult to interpret and it is unclear how the overexpression constructs relate to endogenous localization. If endogenous labeling could be performed, this would be ideal, but at the very least the authors should provide higher resolution images of the data in hand.

We appreciate this comment from the reviewer. The endogenous localization of Rab proteins in sheaths is a difficult experiment to interpret. Rab proteins are abundant within all cells, including axons. If we stained spinal cord tissue and found Rab5+ puncta in sheaths, we would not have the resolution to determine if the proteins were in the sheath or the axon. To better represent the data we have collected, we have included a set of larger insets for our representative images along with 3D reconstructions from Imaris (Figure 7C).

4. The authors should double check the statistical methods employed in Figure 7 and clarify statistics throughout the manuscript to better describe the results.

We appreciate the reviewer’s comment here. We have double checked our statistics and the p-values are correct as presented. We include all of the statistical tests that we perform in each figure legend, and we discuss the use of these tests in the methods. We have submitted all our raw data in spreadsheets to be published with our manuscript.

Reviewer #1 (Recommendations for the authors):There are a number of issues that should be addressed to strengthen the manuscript as currently written.1. Based on the data provided the studies are heavily overinterpreted and the language suggests a more profound result than is actually presented. It is not clear that the retraction of processes represents a destabilization of a complex sheath structure, and the text should be adjusted accordingly.

We greatly appreciate this feed-back and we believe that the reviewer has made a good point. We have now included some definitions at the start of the manuscript (lines 50-69) so that we can make it absolutely clear what we mean with our terminology. We draw your attention to the use of the word “ensheathment”. To ensheathe means “to surround” and in our context an ensheathment is the surrounding of the axon with oligodendrocyte membrane. It is true that we do not know the composition of the oligodendrocyte membrane at this stage of the process. Additionally, the word “destabilized” is only meant to say that the ensheathment was lost from the axon. We have avoided using the term “retract” because we do not want to assume that we know the mechanism for how the oligodendrocyte membrane is disappearing. We have adjusted the text throughout the manuscript to be clearer about what we mean.

2. Along the same lines, OPCs are highly motile and time-lapse studies of oligodendrocytes in vitro show extensive process extension and retraction: it seems likely the authors are seeing the same thing in vivo. It might be interesting to see if transient disruption of actin polymerization or microtubule assembly affected the nature of these early interactions.

This is a great point, and as we note in an earlier comment, we plan to pursue this in the future.

3. The repeated extension/retractions may reflect maturation of the oligodendrocytes and the random association of the nearest target axon to the process tip rather than the proposed molecular tuning. It was not clear the same axon was always targeted. While the data in Figure 2 shows the same axon being contacted, it is less clear in the other figures that it is always the exact same axon. The authors should clarify this point.

Addressed in essential revisions comments above.

4. It is not clear how perturbation of Rab5 results in a loss of sheath stabilization, and Figure 6A seems redundant. This component of the paper seems somewhat out of line with the rest of the manuscript and maybe some additional studies – or even enhanced discussion would help with unification.

We agree that the intention and rationale for looking at Rab5 was not clear. We have modified the text throughout the manuscript to make this more obvious. How Rab5 disruption results in the loss of sheath stabilization is specifically discussed on lines 363-374.

5. The studies would be strengthened by some perturbation studies. Given the short time frame of the initial cell interaction studies would TTX treatment be feasible to see if there is a role for activity in the selection process?

Addressed in essential revisions comments above.

6. Does the difference in sheath number between dorsal and ventral cells reflect axon diameter with more smaller sheaths generated in dorsal regions, or is it a reflection of different populations of oligodendrocytes? Some additional quantification or at least discussion would be helpful.

This is a good question and we have addressed the part about axon diameter more directly in the discussion (lines 375-386).

Reviewer #2 (Recommendations for the authors):There is a wealth of data underpinning this manuscript, particularly with respect to the time-lapse datasets of myelination by oligodendrocytes in the zebrafish spinal cord. Although much of the data align well with previous analyses in the system, some novel insights are documented. The principal new finding is that of the repetitive ensheathment and unwrapping of early myelinating processes from single axons, and the apparently very high rate at which this occurs. I am, however, a little unclear as to how the various analyses in the manuscript relate to one another and think that a few aspects need to be clarified to allow the clearest conclusions to be drawn.1. In Figure 2, there are very nice reconstructions of early ensheathing processes contacting and at least partly enwrapping axons, and exhibiting this wrapping and unwrapping type behaviour. However, what is not clear to me from the analysis presented is to what extent the unwrapping and re-wrapping represents a complete withdrawal from the axon or simply an unwrapping while retaining contact with the axon. It appears from the images shown that this wrapping-unwrapping-wrapping appears to show the myelinating process remains in continuous contact with the axon. One would suspect that the cell biology underlying a local wrapping-unwrapping-wrapping phenomenon, while maintaining contact with the axon to be quite distinct from making contact, wrapping, unwrapping, completely withdrawing from the axon, re-extending a process, re-wrapping etc. The data are clearly to hand, and I think an even more detailed description of the phenomena could be presented. It could obviously be that some processes do the former and others the latter, in which case it would be good to know the relative proportion doing so. This seems like an important additional analysis to me.

Addressed in essential revisions comments above.

2. A huge amount of data centres on the analysis of time-lapse datasets where only the oligodendrocytes are labelled. I realise that this means that the authors can't be as definitive about the wrapping-unwrapping-wrapping of individual axons, but they could presumably tell which ensheathment events happen in the same region very locally, and which involve clear extension of processes and clear retraction of ensheathments from the axon entirely. It is basically the same point as above, but I think it would be very useful to categorise these very dynamic events further so that the field could better understand if we were observing dynamics at the level of processes in contact with axons/ very close to axons, or dynamic events that involved process extension and retraction as well. I hope this my point on this distinction is both clear to the authors and that they see that it will be important to deconstruct to fully appreciate the dynamics of myelination being described.

Addressed in essential revisions comments above.

3. The data regarding the rab manipulations are interesting, but are clearly somewhat disconnected from the very early dynamics of initial ensheathment that represent arguably the most novel insights. I think that the authors could be a little clearer (relates to points 1 and 2) to make it very clear when during the entire process of myelination the rab effects are pronounced. This is reasonably clear when one digs into the data, but the abstract could be read in such a way that one might think that the various uses of the word stabilisation were of one event, where the rab manipulations made here are likely affecting a later stage than the initial part of the paper. Indeed, it could be that the expression of the constructs using a myrf sequence might have made it very hard to find a phenotype at the earlier initial ensheathment phase, so there is a chance of this being a false negative result. Could an alternative driver (olig1?) be used to test one of the candidates to better match the first part of the manuscript. The authors do discuss the limitations of the manipulation, but perhaps this could be a revision experiment worth the effort to have a more streamlined dataset.

We agree that the intention and rationale for looking at Rab5 was not clear. We have modified the text throughout to make this more obvious.

Reviewer #3 (Recommendations for the authors):I suggest the following major experiments:1) Quantify axonal diameter (from Figure 2) to test whether this biases sampling and ultimately stabilization of myelin sheaths.

Addressed in essential revisions comments above.

2) Similarly, test for effect of neuronal activity and/or subtype in early sampling and stabilization. This could be done in a number of ways. For instance, in Figure 2, distinguishing reticulospinal (phox2b+, activity dependent) or CoPA (tbx16+, activity independent) axons could permit measuring propensity for ensheathment attempt and/or stabilization. Alternatively, neuronal activity could be broadly disrupted or enhanced pharmacologically to test for this effect.

Addressed in essential revisions comments above.

3) Given that already such a small proportion of nascent sheaths are stabilized in wild-type, what happens in larvae with a high number of (either mosaic or stably expressed) Rab5-DN oligodendrocytes? Is there ultimately reduced myelination overall, or is the relatively small decrease (per oligodendrocyte) negligible? This would contextualize the importance of endocytic recycling in myelination.

This is a great point. We are planning to perform these experiments in the future.

4) To get at the "why" repeated sampling occurs, the accumulation of synaptic vesicles with (or without) repeated sampling seems an obvious candidate to test. The fact that most destabilized sheaths only sampled the axonal domain once suggests that is insufficient to recruit pro-stabilization factors to the domain, but that 2-4 more sampling attempts could promote SV accumulation.

This is a great point. We are planning to perform these experiments in the future.